# High Variability of Fungal Communities Associated with the Functional Tissues and Rhizosphere Soil of *Picea abies* in the Southern Baltics

**Adas Marčiulynas** [1,*], **Diana Marčiulynienė** [1], **Valeriia Mishcherikova** [1], **Iva Franić** [2], **Jūratė Lynikienė** [1], **Artūras Gedminas** [1] and **Audrius Menkis** [3]

1   Institute of Forestry, Lithuanian Research Centre for Agriculture and Forestry, Liepų Str. 1, LT-53101 Girionys, Lithuania; diana.marciulyniene@lammc.lt (D.M.); valeriia.mishcherikova@lammc.lt (V.M.); jurate.lynikiene@lammc.lt (J.L.); arturas.gedminas@lammc.lt (A.G.)
2   Southern Swedish Forest Research Centre, Swedish University of Agricultural Sciences, Sundsvägen 3, SE-23053 Alnarp, Sweden; iva.franic@slu.se
3   Department of Forest Mycology and Plant Pathology, Uppsala BioCenter, Swedish University of Agricultural Sciences, P.O. Box 7026, SE-75007 Uppsala, Sweden; audrius.menkis@slu.se
*   Correspondence: adas.marciulynas@lammc.lt

**Abstract:** Climate change, which leads to higher temperatures, droughts, and storms, is expected to have a strong effect on both health of forest trees and associated biodiversity. The aim of this study was to investigate the diversity and composition of fungal communities associated with the functional tissues and rhizosphere soil of healthy-looking *Picea abies* to better understand these fungal communities and their potential effect on tree health in the process of climate change. The study sites included 30 *P. abies* stands, where needles, shoots, roots, and the rhizosphere soil was sampled. DNA was isolated from individual samples, amplified using ITS2 rRNA as a marker and subjected to high-throughput sequencing. The sequence analysis showed the presence of 232,547 high-quality reads, which following clustering were found to represent 2701 non-singleton fungal OTUs. The highest absolute richness of fungal OTUs was in the soil (1895), then in the needles (1049) and shoots (1002), and the lowest was in the roots (641). The overall fungal community was composed of Ascomycota (58.3%), Basidiomycota (37.2%), Zygomycota (2.5%), Chytridiomycota (1.6%), and Glomeromycota (0.4%). The most common fungi based on sequence read abundance were *Aspergillus pseudoglaucus* (7.9%), *Archaeorhizomyces* sp. (3.6%), and *Rhinocladiella* sp. (2.0%). Pathogens were relatively rare, among which the most common were *Phacidium lacerum* (1.7%), *Cyphellophora sessilis* (1.4%), and *Rhizosphaera kalkhoffii* (1.4%). The results showed that the detected diversity of fungal OTUs was generally high, but their relative abundance varied greatly among different study sites, thereby highlighting the complexity of interactions among the host trees, fungi, and local environmental conditions.

**Keywords:** biodiversity; climate change; fungi; Norway spruce; pathogens; tree health

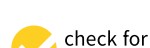



## 1. Introduction

Norway spruce (*Picea abies*) is one of the dominant coniferous tree species of north temperate and boreal forests of Europe, and is therefore of tremendous socio-economic importance [1]. Although it can grow under different climatic and edaphic conditions, it is more adapted to cooler climate and most often found on acidic and nutrient-rich soils with a good availability of moisture [2,3]. As *P. abies* increasingly suffers from different abiotic and biotic damages [4], climate change can be expected to have a major effect on health and distribution range of its forest stands [5,6]. These changes are expected to be due to its sensitivity to changes in the main limiting climatic factors [7,8]. Indeed, climate change modelling shows that the southwestern border of the *P. abies* distribution in Europe at the

end of this century will coincide with the southern border of boreal vegetation zone as it is known today [9,10]. Therefore, *P. abies* stands at or outside this zone can be more prone to damages, i.e., something similar what is currently observed in several Central European countries [11]. These damages are likely to be not only due to the limited adaptation of *P. abies* to changing environmental conditions, but also due to both competition with other newly established tree species and damages caused by indigenous and/or new (invasive) pests and pathogens [12,13]. Interestingly, Bebber et al. [14] showed that in the Northern hemisphere, fungal pathogens expand northwards at a speed of 7.61 ± 2.41 km/year, while physical effects of climate change expand at about 2.7 km/year. Moreover, climate change may also alter the survival and infectivity of fungal pathogens at the same time increasing the susceptibility of host trees, thereby increasing the risk to disease outbreaks. In Northern Europe, the predicted increase of precipitation may favour waterborne pathogens, such as *Phytophthora*, which can be expected to become more frequent in the future [15]. It may also favour outbreaks of fungal pathogens, which benefit from specific climatic conditions and tree stress. For example, *Diplodia sapinea* attacks trees subjected to drought stress [16], while outbreaks of *Gremmeniella abietina* occur after cool and wet summers [17]. Moreover, climate change may lead to the emergence of new virulent strains of fungal pathogens [18]. Several studies on fungal diseases and climate research have revealed that the number of fungal diseases has increased along with climate change and is increasingly recognised as a global threat to important plants [19].

Many different organisms are associated with *P. abies* [3], but these may also be threatened due to changes in habitat quality and availability. Among these are fungi, which play important roles in forest ecosystems, including nutrient and carbon cycles and may have a significant effect on forest health and sustainability [20,21]. Therefore, fungi are known to have a versatile impact on functioning of both individual trees and entire forest ecosystems [22]. The immense diversity of fungal communities across the landscape, however, is due in part to their extensive variability at small spatial scales, something what can be determined by variety of factors such as environmental conditions, plant species identity and diversity, and physical and chemical properties of the soil [23–25]. For example, climatic factors such as precipitation, which can influence the spore germination, and temperature, which can affect the longevity of spores, are important to the distribution and establishment of fungi [26,27]. Soil properties such as pH is known to have a significant effect on fungal diversity including many taxonomic and functional groups [28]. Moreover, changes in climatic factors may lead to the replacement of native fungal species by new (invasive) species with a greater host range and resilience to climate change [29].

*Picea abies* is known to be associated with a diverse fungal community [30–32]. Among the most abundant fungi are endophytes, which are ubiquitous in nature and can be found in different tree tissues [33]. Fungal endophytes are generally defined as species that inhabit tissues without causing apparent disease symptoms [22,34]. By contrast, pathogenic fungi may cause tree diseases, resulting in reduced growth or even mortality. They often attack trees, which are affected by other biotic or/and abiotic factors (see above). Dead trees or their dead tissues are often inhabited by fungal saprotrophs, which obtain nutrients by degrading dead organic matter. Saprotrophs include principal decomposers of tree litter and the wood [35,36]. Ectomycorrhizal fungi, which are specialised soil fungi, form beneficial symbioses with tree roots and can be essential for tree growth and nutrition, particularly under harsh environmental conditions [37]. Changes in diversity and composition of fungal communities may often depend not only on environmental conditions, but also on the health and vitality of host trees [38]. On the other hand, fungal communities may also have a major effect on tree health and require further attention. Majority of previous studies on fungal communities associated with *P. abies* were largely limited to a particular part of the tree such as the phyllosphere [32,39], soil or roots [40,41]. The detection of fungal diversity could also have been limited by some methodological constraints such as, e.g., fungal culturing. The recent development of high-throughput sequencing methods provides powerful tools to explore fungal diversity directly from

environmental samples. Moreover, it generates semiquantitative information and enables taxonomic identification to the species of higher taxonomic level.

The aim of this study was using a holistic approach to investigate the diversity and composition of fungal communities associated with the functional tissues and rhizosphere soil of healthy-looking, but growing under different edaphic and environmental conditions, *P. abies* stands in Lithuania. This was expected to provide a better understanding about these fungal communities and their potential effect on tree health in the process of climate change. The generated knowledge was also expected to be of practical importance as due to climate change, the territory of Lithuania is predicted to be outside the range of *P. abies* distribution by the end of this century [9,10]. We hypothesized that across the study sites, fungal communities associated with *P. abies* are highly variable due to tissue- and site-specific conditions. Specifically, the diversity of fungal communities in the rhizosphere soil and in tree roots, including ectomycorrhizal (ECM) fungi, are dependent on soil chemical parameters, the forest type, and tree age as forest soils represent a highly heterogenous environment and fungal diversity accumulates over years. ECM diversity is largely driven by soil fertility as their importance for tree nutrition decreases with the increase of soil fertility, leading to the shift in community composition as certain ECM fungi prefer more fertile soils than others. Fungal communities in needles and shoots are similar to each other due to more homogeneous habitats and their proximity, and the dependence on climatic factors. Fungal communities in needles and shoots have a lower species diversity as compared to belowground fungal communities as newly produced needles and shoots are colonised each year after their emergence.

## 2. Material and Methods

### 2.1. Study Sites and Sampling

The study sites were at 30 *P. abies* stands distributed throughout the territory of Lithuania (Figure 1, Table 1). These sites were at the same positions as the plots of the Forest Monitoring Level I transnational grid [42], which is used for regular monitoring of stand health, growth, and changes in the forest composition and cover. These sites represent a systematic grid across the country and include a diversity of stand (e.g., age, stand composition, forest site type and vegetation type) and environmental conditions. At each study site, the health condition of *P. abies* trees was assessed using tree damage categories [43]. Meteorological data were obtained from the nearest meteorological stations. Information on stand characteristics and materials sampled (needles, shoots, roots, and the soil) is in Table 1. The classification of forest site type (Table 1) is based on [44]. The classification describes three components, namely the soil moisture, fertility, and granulometric composition, which are indexed as a combination of three letters (one for each component) as, e.g., Ncl. The first letter refers to soil moisture: N—soils of normal atmospheric moisture, groundwater is usually deeper than 3 m from the surface; L—temporary water-logged and gleyic soils; P—non-drained forest wetlands (characterised based on the thickness of the peat layer). The second letter of the index describes soil fertility: a—very infertile; b—poor fertility; c—moderate fertility; d—high fertility. The third letter of the index shows the granulometric composition: l—light soils (sand, sandy loam, and gravel); s—heavy soils (loam, clay, chalk, dolomite, gypsum); p—binary soils, when a layer of the light soil is on the heavy one (deposited deeper than 50 cm from the surface). Vegetation typology is based on [45], which describes the composition of phytocenosis, i.e., the composition of forest stand, shrubs, grasses, mosses, forest stand productivity, and habitat conditions, using Latin names of grasses and mosses (Table 1). Although those species, which are used for naming, are not always dominant on a particular site, the vegetation type is identified based on the characteristics of the whole vegetation.

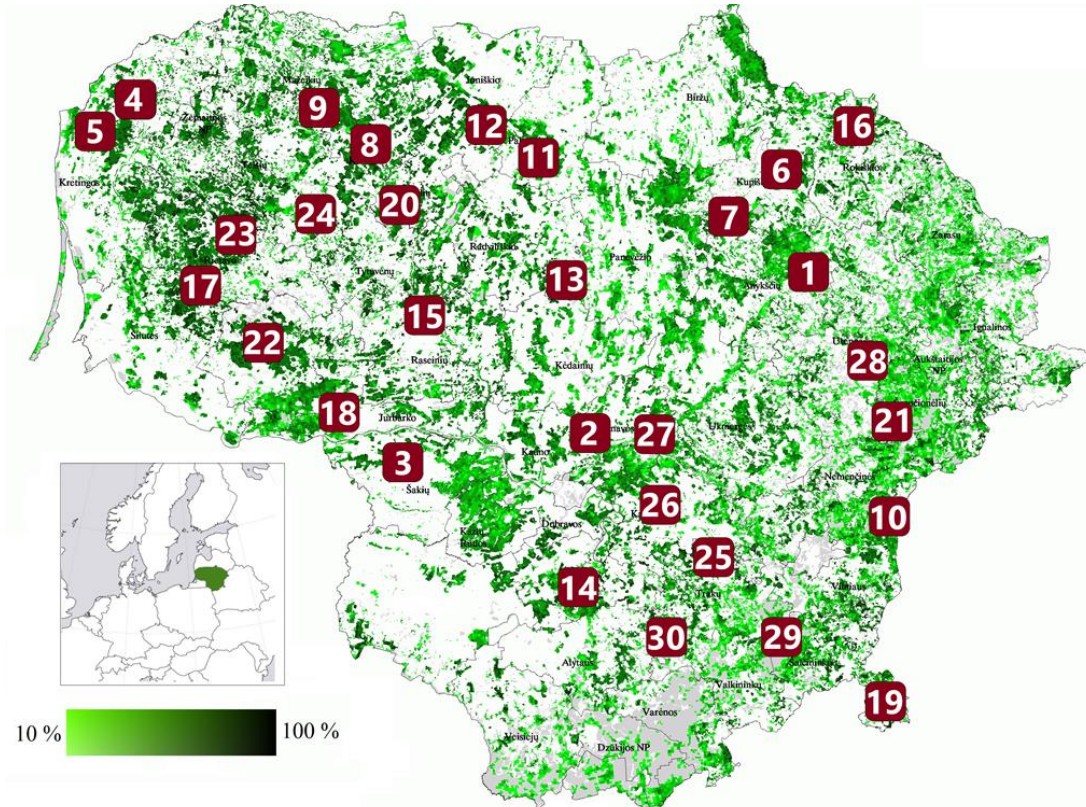

**Figure 1.** Map of Lithuania (position shown on the north European map in the lower left corner) showing the distribution of Norway spruce (*Picea abies*) forest stands (in green), where sampling of living shoots, needles, roots, and the rhizosphere soil was carried out. The gradient bar shows the percentage of *P. abies* trees in the composition of forest stands.

At each site, for sampling of soil, the litter layer was removed, and samples were taken in the vicinity of *P. abies* trees down to 20 cm depth using a 2 cm diameter soil core, which was carefully cleaned between individual samples. Soil samples included five random replicates per site. Each soil sample consisted of ca. 50 g of organic soil layer and ca. 50 g of mineral soil layer. In total, 150 soil samples were collected. Samples of fine roots were excavated in the vicinity of five random *P. abies* trees. The soil was removed, and each sample included up to seven fine roots with root tips (up to 10 g in total). In total, 150 root samples were collected. Shoot and needle samples were taken from ten random *P. abies* trees. Telescopic secateurs were used to cut 2-year-old shoots with needles from the middle part of crowns (about 10–12 m above the ground). An individual needle sample (one per tree) consisted of 25 healthy-looking needles, which were randomly collected from cut shoots using forceps, which were cleaned between individual samples. Shoot samples were prepared by removing remaining needles and cutting them into ca. 5 cm segments. In total, 300 needle and 300 shoot samples were collected. Individual soil, root, shoot, and needle samples were placed separately into plastic bags and labelled. The same day of sampling, samples were transported to the laboratory and placed in $-20\,^{\circ}\mathrm{C}$ for storage.

To determine the chemical and physical properties of the soil, five random soil samples (ca. 200 g) per site were taken in the vicinity of *P. abies* trees and pooled together. In the laboratory, the collected soil samples were sieved using a $2 \times 2$ mm sieve to separate fine fraction soil. The pH of the soil was determined in the KCl extract using the potentiometric method (ISO 10390:2005), available phosphorus ($P_2O_5$), and potassium $K_2O$ (K) (mg kg$^{-1}$ soil) using the Egner–Rim–Doming (A-L) method.

**Table 1.** Characteristics of *Picea abies* forest stands where needles, shoots, roots, and the rhizosphere soil were sampled.

| Site No. | Geographical Position | | Stand Characteristics | | | | Soil Chemical Parameters | | | | | | | Meteorological Data | | Stand Sanitary Condition | | |
|---|---|---|---|---|---|---|---|---|---|---|---|---|---|---|---|---|---|---|
| | N | E | Forest [a]/ Vegetation Type [b] | Tree Species Composition, % [c] | Age (y) | pH (KCl), mol/L | $P_2O_5$, mg/kg | $K_2O$, mg/kg | Ca, mg/kg | Mg, mg/kg | Cl, mg/kg | Salts, ms/cm | Average Temp., °C | Precipitation, mm/year | Defoliation, % | Dechromation, % | Dry Branches, % |
| 1 | 55°37′ | 25°11′ | Ncl/ox | 100S | 50 | 3.9 | 306 | 43 | 267 | 72 | 3.6 | 3.35 | 8.0 | 508.8 | 11.2 | 5.8 | 14.5 |
| 2 | 55°01′ | 24°12′ | Lcl/mox | 100S | 52 | 3.3 | 56 | 39 | 242 | 56 | 5.3 | 4.33 | 8.3 | 614.9 | 29.0 | 13.5 | 26.0 |
| 3 | 54°52′ | 23°26′ | Pcn/fils | 100S | 61 | 2.4 | 65 | 296 | 774 | 279 | 8.9 | 11.7 | 8.6 | 519.8 | 13.2 | 2.3 | 12.8 |
| 4 | 56°12′ | 21°27′ | Lbp/m | 90S, 10P | 59 | 3.0 | 32 | 170 | 583 | 136 | 5.3 | 7.64 | 8.9 | 491.7 | 15.8 | 4.1 | 13.5 |
| 5 | 56°00′ | 21°07′ | Lbl/m | 100S | 59 | 2.8 | 12 | 69 | 264 | 73 | 5.3 | 8.83 | 8.9 | 491.7 | 16.3 | 3.7 | 13.5 |
| 6 | 55°54′ | 25°13′ | Lcs/mox | 90S, 10A | 67 | 5.8 | 15 | 47 | 2256 | 490 | 3.6 | 6.49 | 7.7 | 514.6 | 18.0 | 4.8 | 15.8 |
| 7 | 55°44′ | 24°43′ | Lcl/mox | 90S, 10P | 57 | 3.1 | 8 | 21 | 154 | 28 | 1.8 | 2.36 | 8.1 | 479.8 | 24.8 | 7.0 | 20.7 |
| 8 | 56°03′ | 22°56′ | Lds/oxn | 100S | 48 | 5.1 | 12 | 38 | 1472 | 329 | 5.3 | 13.6 | 8.1 | 505.5 | 16.5 | 6.8 | 14.7 |
| 9 | 56°11′ | 22°23′ | Lcp/mox | 90S, 10P | 58 | 5.2 | 16 | 144 | 1770 | 270 | 5.3 | 4.87 | 8.0 | 480.4 | 18.5 | 4.0 | 17.5 |
| 10 | 54°52′ | 25°42′ | Ncl/ox | 100S | 35 | 3.8 | 169 | 40 | 176 | 48 | 3.6 | 2.01 | 8.0 | 755.4 | 18.3 | 6.2 | 14.3 |
| 11 | 55°53′ | 24°12′ | Ldf/oxn | 80S, 20F | 62 | 5.7 | 17 | 75 | 2968 | 630 | 5.3 | 12.4 | 8.1 | 505.5 | 16.7 | 8.0 | 17.0 |
| 12 | 56°03′ | 23°40′ | Lds/oxn | 90S, 10A | 42 | 4.4 | 11 | 131 | 2952 | 542 | 7.1 | 26.0 | 8.1 | 505.5 | 9.0 | 1.5 | 11.2 |
| 13 | 55°29′ | 24°12′ | Lds/oxn | 100S | 53 | 4.8 | 25 | 76 | 1294 | 197 | 5.3 | 4.7 | 8.1 | 479.8 | 12.7 | 3.7 | 9.3 |
| 14 | 54°35′ | 23°57′ | Ncl/ox | 100S | 47 | 3.9 | 27 | 21 | 101 | 32 | 7.1 | 2.55 | 8.6 | 595.2 | 8.8 | 3.3 | 13.8 |
| 15 | 55°27′ | 23°27′ | Nds/hox | 100S | 50 | 4.0 | 19 | 77 | 942 | 147 | 5.3 | 3.42 | 7.8 | 458 | 18.5 | 5.3 | 14.8 |
| 16 | 56°03′ | 25°42′ | Lcp/mox | 60S, 30PT, 10B | 60 | 3.7 | 36 | 63 | 286 | 73 | 3.6 | 3.14 | 7.7 | 514.6 | 18.3 | 7.2 | 15.3 |
| 17 | 55°28′ | 21°57′ | Lcp/mox | 60S, 20B, 10Q, 10A | 59 | 3.8 | 11 | 107 | 791 | 122 | 3.6 | 4.95 | 8.4 | 617.0 | 14.3 | 3.7 | 12.2 |
| 18 | 55°02′ | 22°41′ | Lcl/mox | 60S, 40P | 23 | 2.8 | 15 | 86 | 277 | 75 | 3.6 | 3.64 | 8.6 | 519.8 | 19.3 | 7.7 | 16.8 |
| 19 | 54°18′ | 25°40′ | Ncl/ox | 80S, 10Q, 10PT | 39 | 3.8 | 71 | 43 | 157 | 42 | 5.3 | 3.06 | 8.0 | 530.6 | 15.8 | 2.0 | 12.7 |
| 20 | 54°18′ | 25°40′ | Nds/hox | 100S | 53 | 5.0 | 27 | 48 | 2584 | 374 | 7.1 | 11.1 | 8.1 | 479.8 | 25.3 | 16.5 | 25.7 |
| 21 | 55°10′ | 25°42′ | Nbl/v | 90S, 10P | 40 | 4.1 | 78 | 36 | 238 | 52 | 5.3 | 3.96 | 7.4 | 669.2 | 18.7 | 8.0 | 13.8 |
| 22 | 55°19′ | 22°27′ | Lcp/mox | 90S, 10B | 55 | 3.3 | 118 | 300 | 2092 | 566 | 7.1 | 18.5 | 8.6 | 591.3 | 8.7 | 0.3 | 12.7 |
| 23 | 55°45′ | 21°41′ | Ncs/ox | 100S | 50 | 4.1 | 71 | 123 | 2043 | 161 | 3.6 | 5.0 | 8.0 | 598.7 | 14.7 | 3.5 | 15.0 |
| 24 | 55°44′ | 22°25′ | Ncl/ox | 100S | 40 | 3.5 | 37 | 25 | 212 | 48 | 3.6 | 2.07 | 8.0 | 598.7 | 16.7 | 6.0 | 16.7 |
| 25 | 54°44′ | 24°42′ | Ncl/ox | 100S | 58 | 3.8 | 12 | 70 | 416 | 136 | 3.6 | 4.65 | 7.3 | 614.9 | 22.0 | 7.8 | 18.3 |

**Table 1.** *Cont.*

| Site No. | Geographical Position | | Stand Characteristics | | | Soil Chemical Parameters | | | | | | | Meteorological Data | | Stand Sanitary Condition | | |
|---|---|---|---|---|---|---|---|---|---|---|---|---|---|---|---|---|---|
| | N | E | Forest [a]/ Vegetation Type [b] | Tree Species Composition, % [c] | Age (y) | pH (KCl), mol/L | $P_2O_5$, mg/kg | $K_2O$, mg/kg | Ca, mg/kg | Mg, mg/kg | Cl, mg/kg | Salts, ms/cm | Average Temp., °C | Precipitation, mm/year | Defoliation, % | Dechromation, % | Dry Branches, % |
| 26 | 54°56′ | 24°42′ | Lbl/m | 80S, 10B, 10P | 60 | 4.7 | 26 | 68 | 2362 | 308 | 5.3 | 14.0 | 8.0 | 535.8 | 13.7 | 2.5 | 10.7 |
| 27 | 55°10′ | 24°41′ | Ncl/ox | 60S, 20P, 20PT | 50 | 6.6 | 544 | 183 | 11,508 | 1184 | 3.6 | 8.43 | 8.0 | 506.9 | 21.3 | 5.7 | 17.2 |
| 28 | 55°18′ | 25°42′ | Pcn/fils | 90S, 10B | 52 | 3.0 | 132 | 237 | 2584 | 520 | 14.2 | 33 | 7.7 | 505.4 | 19.3 | 5.2 | 17.7 |
| 29 | 54°25′ | 24°57′ | Nbl/v | 100S | 68 | 3.4 | 23 | 31 | 161 | 42 | 3.6 | 2.73 | 8.0 | 593.2 | 16.2 | 2.7 | 12.8 |
| 30 | 54°25′ | 24°27′ | Ncl/ox | 70S, 20B, 10P | 48 | 3.7 | 169 | 53 | 512 | 117 | 3.6 | 5.0 | 8.0 | 540.9 | 14.2 | 3.3 | 11.2 |

[a] N: Normal moisture; L: temporary waterlogged soils; P: wetlands. b: poor fertility; c: moderate fertility; d: high fertility. l: light soil texture; s: heavy soils; p: binary soils [44]. [b] ox: *oxalidosum*; oxn: *oxalido-nemoroso-Piceetum*; m: *myrtillosa*; mox: *myrtillo-oxalidosa*; v: *vacciniosa*; hox: *Hepatico-oxalidosa*; fils: *Filipendulo-mixtoherbosa* [45]. [c] S: *Picea abies*; P: *Pinus sylvestris*; B: *Betula pendula*; A: *Alnus incana*; F: *Fraxinus excelsior*; PT: *Populus tremula*. In each stand, tree species composition is based on the volume.

*2.2. DNA Isolation, Amplification, and Sequencing*

The principles of the DNA work followed the study by Marčiulynienė et al. [46]. Prior to the isolation of the DNA, each sample (needles, shoots, roots, and the soil) was freeze-dried using a Labconco FreeZone Benchtop Freeze Dryer (Cole-Parmer, Vernon Hills, IL, USA) at $-60$ °C for two days. After the freeze-drying, ca. 0.03 g dry weight of each needle, shoots or root sample was placed into a 2-mL screw-cap centrifugation tube together with glass beads. No surface sterilization of the samples was carried out. Samples were homogenized using a Fast prep shaker (Bertin Technologies, Montigny-le-Bretonneux, France). The DNA was isolated using CTAB extraction buffer (0.5 M EDTA pH 8.0, 1 M Tris-HCL pH 8.0, 5 M NaCl, 3% CTAB) followed by incubation at 65 °C for 1 h. After the centrifugation, the supernatant was transferred to a new 1.5-mL Eppendorf tube, mixed with an equal volume of chloroform, centrifuged at 10,000 rpm for 8 min, and the upper phase was transferred to new 1.5-mL Eppendorf tubes. Then, an equal volume of 2-propanol was used to precipitate the DNA into a pellet by centrifugation at 13,000 rpm for 20 min. The pellet was washed in 500 μL 70% ethanol, dried, and dissolved in 30 μL sterile milli-Q water. Differently from other samples, ca. 1 g of freeze-dried soil per each sample was used for the isolation of the DNA using a NucleoSpin®Soil kit (Macherey-Nagel GmbH & Co., Düren, Germany) according to the manufacturer's recommendations. Following the isolation, the DNA concentration in individual samples (needles, shoots, roots, and the soil) was determined using a NanoDrop™ One spectrophotometer (Thermo Scientific, Rodchester, NY, USA) and adjusted to 1–10 ng/μL. Amplification of the ITS2 rRNA region was done using a fungal specific primer gITS7 [47] and a universal primer ITS4 [48], both containing sample identification barcodes. Samples of the same substrate (needles, shoots, roots of the soil) and site were amplified using primers with the same barcode. PCR was performed in 50 μL reactions and consisted of the following final concentrations, 0.25 ng/μL-template DNA, 200 μM of dNTPs; 750 μM of $MgCl_2$; 0.025 μM DreamTaq Green polymerase (5 U/μL) (Thermo Scientific, Waltham, MA, USA), and 200 nM of each primer. Amplifications were performed using the Applied Biosystems 2720 thermal cycler. The PCR program started with denaturation at 95 °C for 5 min, followed by 30 cycles of 95 °C for 30 s, annealing at 56 °C for 30 s and 72 °C for 30 s, followed by a final extension step at 72 °C for 7 min. The PCR products were assessed using gel electrophoresis on 1% agarose gel stained with Nancy-520 (Sigma-Aldrich, Stockholm, Sweden). PCR products were purified using 3 M sodium acetate (pH 5.2) (Applichem Gmbh, Darmstadt, Germany) and 96% ethanol mixture (1:2). After quantification of PCR products using a Qubit fluorometer 4.0 (Life Technologies, Stockholm, Sweden), samples were pooled in an equimolar mix and used for PacBio sequencing using two SMRT cells at the SciLifeLab in Uppsala, Sweden.

*2.3. Bioinformatics*

The sequence reads were subjected to control of quality and clustering using the SCATA NGS sequencing pipeline at http://scata.mykopat.slu.se (accessed on 10 September 2021). Quality filtering was done by removing short sequences (<200 bp), sequences with low read quality (Q < 20), primer dimers, and homopolymers, which were collapsed to 3 base pairs (bp) before clustering. Sequences lacking a tag or primer were also removed. The primer and sample barcodes were then removed, but information on the sequence association with the sample was stored as meta-data. The sequences were clustered into different OTUs using single-linkage clustering based on 98% similarity. The most common genotype (real read) for each cluster was used to represent each OTU. For clusters containing two sequences, a consensus sequence was produced. Fungal OTUs were taxonomically identified using the GenBank (NCBI) database and the Blastn algorithm. The criteria used for identification were: sequence coverage >80%; similarity to species level 98–100%, similarity to genus level 94–97%. Sequences not matching these criteria were given unique names. Representative sequences of fungal non-singletons as the Targeted Locus Study project have been deposited in GenBank under accession number KFPS00000000. Fungal functional groups were annotated using the FUNGuild fungal database [49], and, if needed,

were further refined using information at the MycoBank database. In case the fungus had two possible functional groups, it was classified based on the FUNGuild categorisation.

*2.4. Statistical Analyses*

Rarefaction analysis was performed using Analytical Rarefaction v.1.3 available at http://www.uga.edu/strata/software/index.html (accessed on 15 November 2021). Correlation analysis of species richness among different study sites and substrates (needles, shoots, roots, and the soil) was carried out in Minitab v. 18.1 (University Park, Pennsylvania, PA, USA). The Shannon diversity index and qualitative Sørensen similarity index were used to characterise the diversity of fungal communities [50,51]. The nonparametric Mann–Whitney test in Minitab was used to test if the Shannon diversity index among different sites and samples was statistically similar or not. The effects of the substrate, environmental variables at a site level, and soil characteristics at a site level on species richness (number of fungal OTUs per sample) were assessed using generalized linear mixed effect models (glmmTMB function from the glmmTMB package [52]). Correlation between predictor variables was assessed using cor function in R [53]. When the correlation coefficient between the two variables was higher than 0.7, only one variable was selected to be used in the final model. Final model contained the following variables as fixed factors: tree composition, age, and defoliation (i.e., stand variables), average annual temperature and annual precipitation (i.e., environmental variables), and soil pH, $P_2O_5$, $K_2O$, and salts (i.e., soil variables). The interactions between each variable and the substrate were included in the model to assess if the effects of a variable were consistent across all substrates. The site was included in the model as a random factor. All continuous variables were scaled using the scale function in R [53]. Model predictions were calculated using ggpredict from the package ggeffects [54] in R and plotted using the ggplot function from the ggplot2 package [55].

The composition of fungal communities was studied using non-metric multidimensional scaling (NMDS) based on the Bray–Curtis similarity index. Analyses were carried out using both the complete dataset and the dataset without rare (<50 reads) OTUs. One-way ANOSIM was performed to test for significant differences among different substrates. Tukey's method was used for creating a set of confidence intervals between the means. These analyses were performed using Vegan 2.5.7 [56] and Stats 3.6.2 in R [53].

## 3. Results

High-throughput sequencing and quality filtering showed the presence of 232,548 high-quality reads, which following clustering (at 98% similarity level) analysis were found to represent 3016 non-singleton OTUs. Among the non-singletons, 2701 (89.6%) were representing fungi (Supplementary Table S1), while the remaining 315 (10.4%) non-fungal OTUs were excluded. The number of high-quality sequences and fungal OTUs from each study site and substrate is in Table 2.

When all sites were taken together, a plot of fungal OTUs vs. the number of fungal sequences resulted in rarefaction curves, which did not reach the asymptote (Figure 2). When the same number of sequences had been taken from different substrates (needles, shoots, roots, and the soil), the species richness was significantly higher in the soil than in other substrates ($p < 0.05$). Furthermore, the richness of fungal OTUs was significantly higher in the shoots and needles than in the roots ($p < 0.05$). In a similar comparison, shoots and needles did not differ significantly from each other ($p > 0.05$).

Within each substrate (needles, shoots, roots, and the soil), the richness of fungal OTUs varied greatly among individual study sites (Figure 3). Correlation analysis showed that there was no significant correlation when the richness of fungal OTUs was compared among different substrates ($p > 0.05$).

**Table 2.** Generated high-quality fungal sequences and detected diversity of fungal OTUs in different substrates from 30 *Picea abies* forest stands in Lithuania.

| Site | No. of Sequences/Fungal OTUs | | | | Shannon Diversity Index (H) | | | |
|---|---|---|---|---|---|---|---|---|
| No. | Needles | Shoots | Roots | Soil | Needles | Shoots | Roots | Soil |
| 1 | 4222/65 | 636/132 | 3136/145 | 1792/160 | 0.34 | 3.95 | 2.77 | 3.52 |
| 2 | 2533/27 | 425/104 | 413/87 | 4586/295 | 0.14 | 4.01 | 3.27 | 4.38 |
| 3 | 1664/182 | 645/91 | 702/47 | 24/19 | 4.16 | 3.42 | 2.49 | 2.92 |
| 4 | 2605/241 | 562/136 | 139/31 | 1769/310 | 4.36 | 3.43 | 1.68 | 3.82 |
| 5 | 2215/94 | 406/72 | 21/7 | 2327/210 | 1.12 | 4.11 | 2.55 | 4.96 |
| 6 | 678/51 | 1088/120 | 706/79 | 2782/346 | 1.25 | 3.36 | 3.15 | 4.78 |
| 7 | 831/143 | 247/64 | 277/57 | 1079/171 | 4.28 | 3.66 | 3.03 | 3.85 |
| 8 | -/- | 1861/186 | 205/38 | 1104/185 | - | 3.81 | 2.92 | 4.13 |
| 9 | 38/22 | 594/73 | 1086/102 | 2212/192 | 2.89 | 2.87 | 3.22 | 3.00 |
| 10 | 722/132 | 645/117 | 128/39 | 1632/220 | 4.06 | 3.89 | 3.15 | 4.26 |
| 11 | 245/80 | 303/73 | -/- | 2796/338 | 3.67 | 3.44 | - | 4.89 |
| 12 | 1020/139 | 446/75 | 116/41 | 2275/313 | 3.52 | 3.57 | 3.16 | 5.01 |
| 13 | 2676/216 | 3314/145 | 1047/117 | 2011/292 | 3.78 | 3.18 | 3.07 | 4.67 |
| 14 | 2860/203 | 98/41 | 3136/149 | 1295/160 | 4.14 | 3.27 | 3.00 | 4.34 |
| 15 | 7090/300 | 1271/190 | 3/2 | 629/158 | 3.95 | 4.34 | 0.64 | 4.41 |
| 16 | 5675/84 | 109/36 | 60/22 | 2450/209 | 4.20 | 3.01 | 2.35 | 3.72 |
| 17 | 865/150 | 2173/216 | 53/18 | 2899/316 | 1.01 | 4.30 | 2.25 | 4.49 |
| 18 | 6501/316 | 11,512/201 | 273/59 | 119/61 | 4.18 | 3.25 | 3.12 | 3.62 |
| 19 | 2276/232 | 2269/189 | 46/26 | 850/184 | 4.29 | 4.09 | 3.08 | 4.56 |
| 20 | 1127/161 | 1312/100 | 1041/63 | 5301/269 | 4.06 | 2.53 | 2.41 | 3.82 |
| 21 | 1205/149 | 1171/146 | 178/24 | 8092/444 | 3.96 | 3.88 | 2.59 | 3.99 |
| 22 | 5757/102 | 1699/145 | 150/42 | 86/44 | 0.72 | 3.73 | 3.06 | 3.42 |
| 23 | 1447/155 | 3994/270 | 84/18 | 11,181/472 | 4.06 | 4.08 | 1.95 | 4.33 |
| 24 | 2898/202 | 2521/173 | 385/63 | 2354/309 | 4.09 | 3.21 | 3.03 | 4.87 |
| 25 | 2927/208 | 2070/162 | 135/25 | 3830/310 | 4.17 | 4.13 | 2.28 | 4.22 |
| 26 | 821/128 | 1642/103 | 417/51 | 368/80 | 3.89 | 2.99 | 2.68 | 3.73 |
| 27 | 1328/172 | 952/132 | 856/76 | 7218/410 | 4.05 | 3.91 | 3.06 | 4.12 |
| 28 | 58/19 | 6703/226 | 3314/94 | 8181/434 | 2.03 | 3.56 | 2.62 | 4.18 |
| 29 | 259/68 | 7618/242 | 1539/116 | 898/121 | 3.21 | 3.54 | 3.29 | 3.95 |
| 30 | 423/104 | 1802/192 | 807/62 | 6811/273 | 3.89 | 4.05 | 2.60 | 3.02 |
| Total | 62,966/1049 | 60,188/1002 | 20,443/641 | 88,951/1895 | | | | |

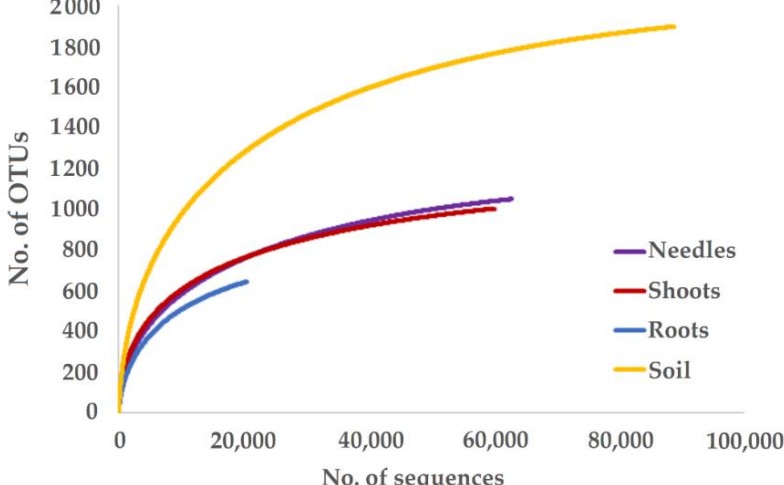

**Figure 2.** Rarefaction curves showing the relationship between the cumulative number of fungal OTUs and the number of ITS rRNA sequences in needles, shoots, roots, and the rhizosphere soil.

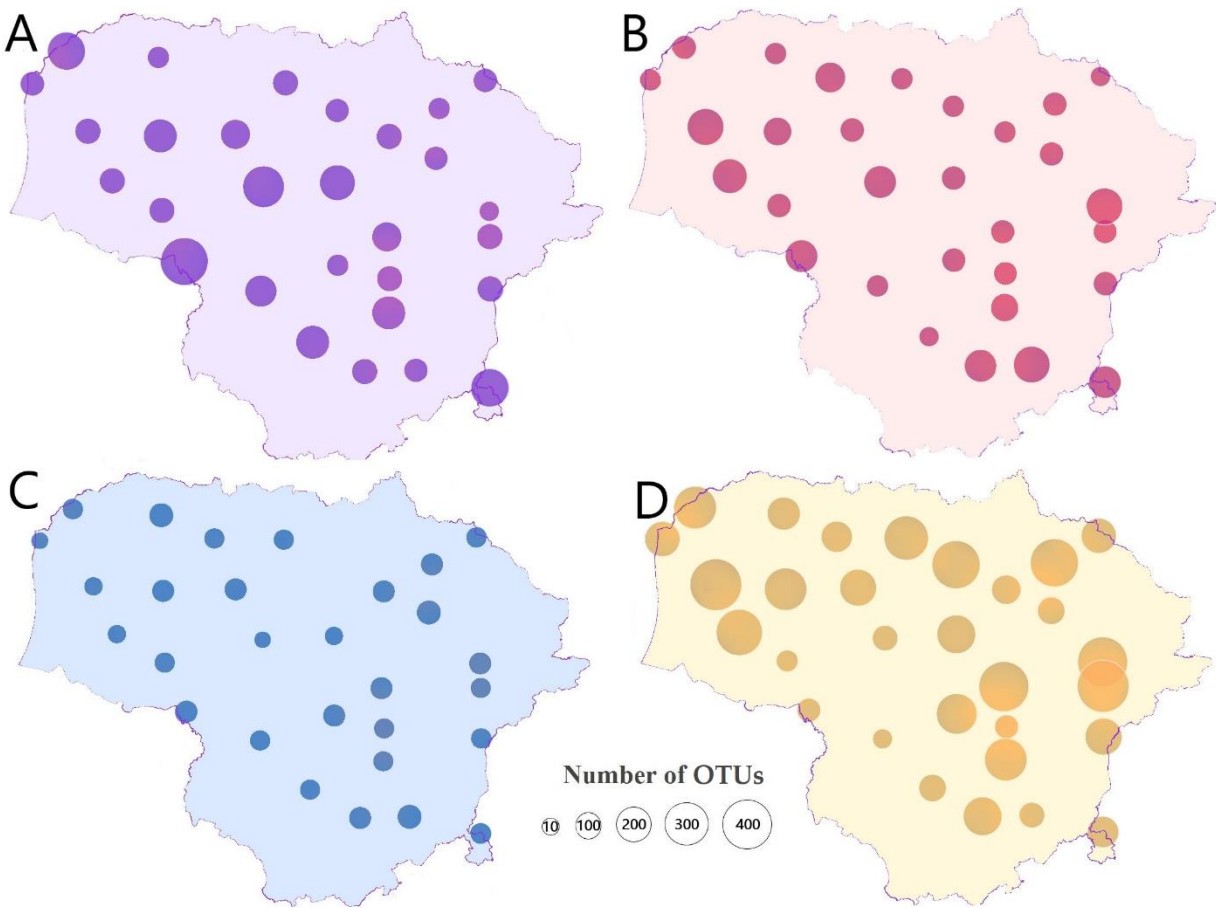

**Figure 3.** Map of Lithuania showing the richness of fungal OTUs associated with (**A**) needles, (**B**) shoots, (**C**) roots, and (**D**) the rhizosphere soil at different *Picea abies* sampling sites.

By contrast, the species richness was significantly affected by variables reflecting the soil chemistry. The pH had a similar effect on shoot and root fungi—species richness decreased with increased pH. The opposite trend was for soil fungi, while species richness of fungi in needles remained generally unaffected ($\chi^2 = 73$, df = 3, $p < 0.05$) (Figure 4A). The increase of species richness with increased $P_2O_5$ was only in roots, while fungi in all other substrates showed the decrease of species richness with the increasing $P_2O_5$ values ($\chi^2 = 108$, df = 3, $p < 0.05$) (Figure 4B). The increasing concentration of $K_2O$ increased species richness in needles, but decreased in roots and soil, while fungi in shoots were generally unaffected ($\chi^2 = 252$, df = 3, $p < 0.05$) (Figure 4C). The species richness increased in root and in the soil with the increasing concentration of salts, while the opposite was for needle and shoot fungi ($\chi^2 = 156$, df = 3, $p < 0.05$) (Figure 4D).

The fungal species richness in roots increased with the stand age, while the opposite was for fungi in needles and the soil. The age of the stand did not affect the species richness in shoots ($\chi^2 = 280$, df = 3, $p < 0.05$) (Figure 4E). The increasing defoliation increased fungal species richness in needles and the soil, while the opposite was in shoots and roots ($\chi^2 = 96$, df = 3, $p < 0.05$) (Figure 4F). Climatic variables also showed a significant effect on species richness that varied depending on the type of substrate ($\chi^2 = 34$, df = 3, $p < 0.05$ and $\chi^2 = 191$, df = 3, $p < 0.05$ for temperature and precipitation, respectively). The fungal species richness in needles was decreasing with the increase of temperature and precipitation. In shoots, the fungal species richness decreased with the increase of temperature but increased with the increase of precipitation. The fungal species richness in roots and soil increased with the increase of precipitation. Although increasing temperature had a positive effect on richness of root fungi, the opposite was for fungi in the soil (Figure 4G,H).

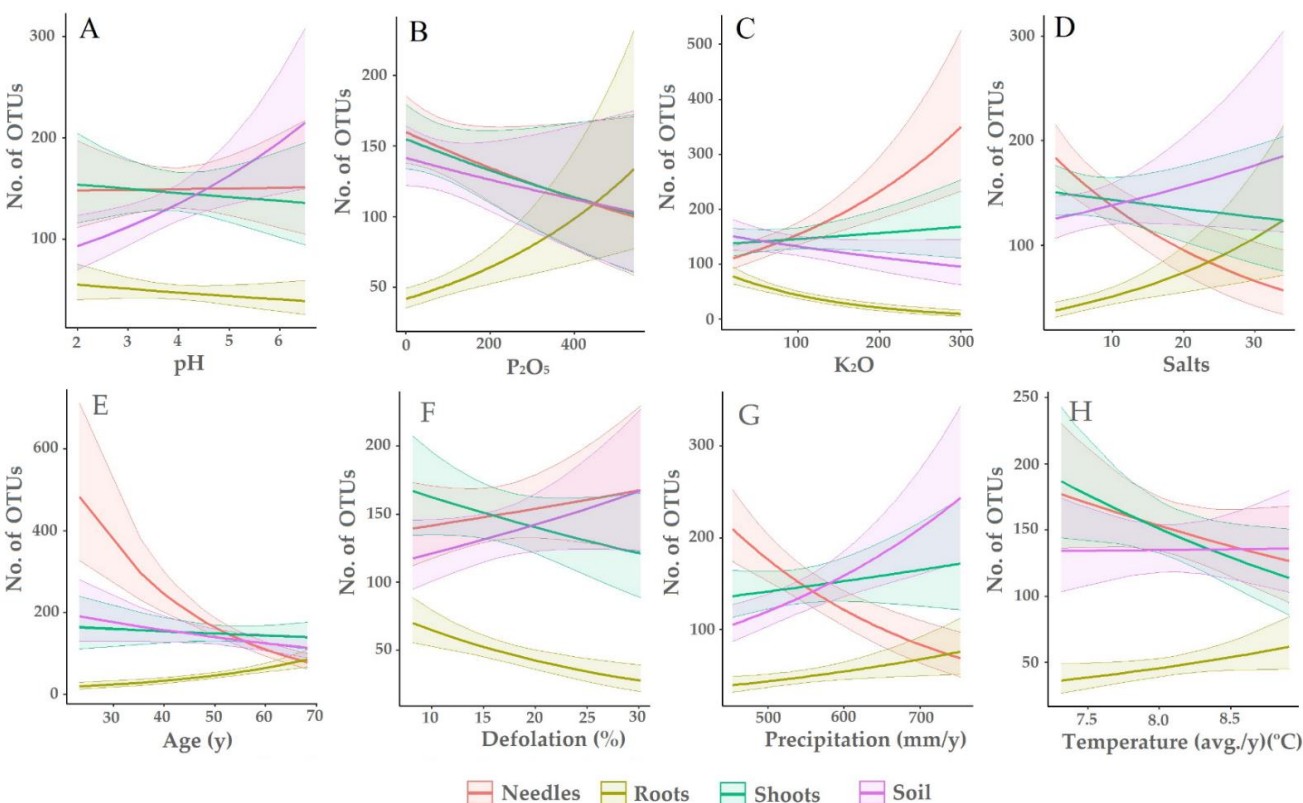

**Figure 4.** The relationship between the richness of fungal OTUs in different substrates (needles, shoots, roots, and the soil) of *Picea abies* and soil pH (**A**), $P_2O_5$ (**B**), $K_2O$ (**C**), salts (**D**), stand age (**E**), tree defoliation (**F**), yearly precipitation (**G**) and average yearly temperature (**H**). The semitransparent field around each curve denotes the size of deviation from the mean value.

Among all fungal OTUs, 202 (7.5%) were exclusively found in shoots, 167 (6.2%)—in roots, 188 (6.9%)—in needles, 920 (34.1%)—in the soil, and 125 (4.6%) were shared among different substrates (Figure 5). The lowest number of shared OTUs was between the root and shoot samples (11), while the highest number was between the root and soil samples (232) (Figure 5).

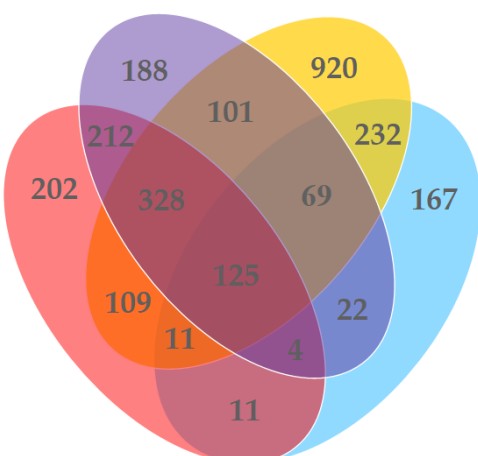

**Figure 5.** Venn diagram showing the diversity and overlap of fungal OTUs in different substrates, collected in *Picea abies* stands. Different colours represent different substrates: violet—needles, red—shoots, blue—roots, and yellow—soil.

Taxonomic identification showed that among all samples, Ascomycota accounted for 1575 (58.3%) fungal OTUs, followed by 1005 (37.2%) OTUs of Basidiomycota, 68 (2.5%) of Zygomycota, 42 (1.6%) of Chytridiomycota, 10 (0.4%) of Glomeromycota, and the least common was Neocallimastigomycota, which included one (0.04%) OTU.

The distribution and relative abundance of fungal classes varied among different sites and substrates (Figure 6). Among all sites, the most dominant fungal classes in shoots were Dothideomycetes (33.6%), Eurotiomycetes (12.8%), and Tremellomycetes (12.2%). In the needles, these were Sordariomycetes (31.0%) and Dothideomycetes (18.0%), in the roots—Agaricomycetes (37.6%), Leotiomycetes (24.8%), and Archaeorhizomycetes (20.0%), while in the soil—Dothideomycetes (22.8%), Agaricomycetes (22.2%), and Leotiomycetes (15.0%) (Figure 6).

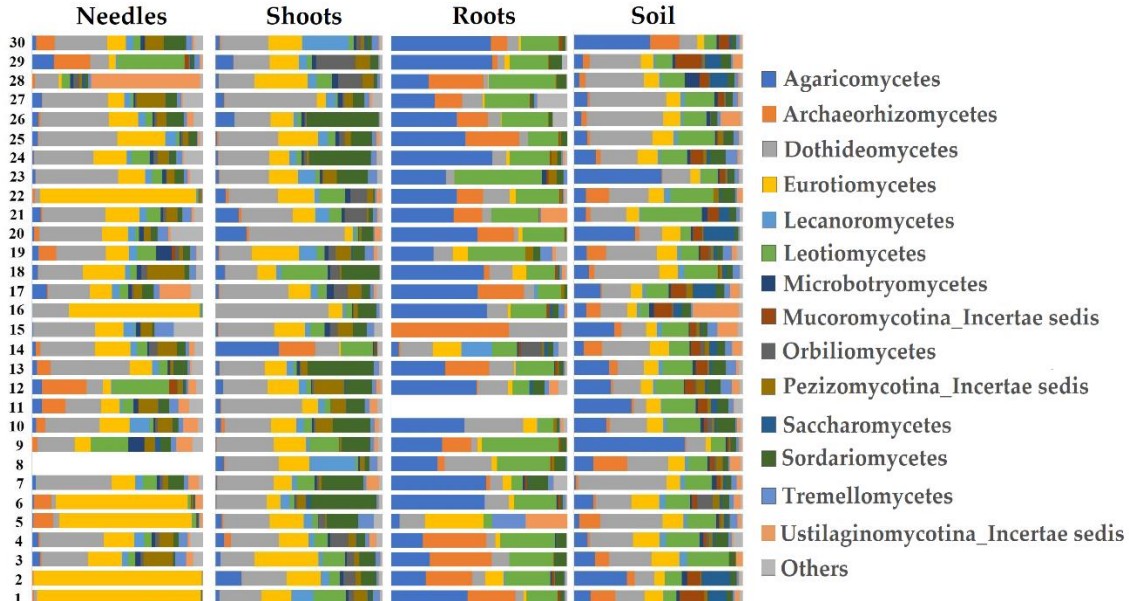

**Figure 6.** Relative abundance (%) of fungal classes in needles, shoots, roots, and the rhizosphere soil of *Picea abies*. *Others* denote fungal classes with a relative abundance of <1%. Site numbers to the left are as in Table 1 and Figure 1.

The most common fungal OTUs in shoots were *Arachnopeziza* sp. 5208_27 (6.0% of all high-quality sequences), *Rhinocladiella* sp. 5208_3 (5.4%), and *Lophium arboricola* (4.6%), in needles—*Aspergillus pseudoglaucus* (29.9%), *Cyphellophora sessilis* (3.8%), *Rhizosphaera kalkhoffii* (3.5%) and *Trichomerium* sp. 5208_23 (2.7%), in roots—*Archaeorhizomyces* sp. 5208_0 (18.4%), *Phialocephala fortinii* (9.8%), *Mycena cinerella* (9.5%) and *Trechispora* sp. 5208_19 (9.4%), and in the soil, these were *Phacidium lacerum* (3.7%), *Agaricomycetes* sp. 5208_12 (3.4%), and *Archaeorhizomyces* sp. 5208_0 (3.3%) (Table 3).

**Table 3.** Occurrence and relative abundance of the 25 most common fungi (shown as a proportion of all high-quality fungal sequences) found in different substrates. The data from different sites are combined.

| Fungal OTUs | Phylum * | Genbank Reference | Similarity, % | Needles, % | Shoots, % | Roots, % | Soil, % | All, % | Ecological Role |
|---|---|---|---|---|---|---|---|---|---|
| *Aspergillus pseudoglaucus* | A | MT582752 | 100 | 29.9 | 0.01 | 0.1 | 0.05 | 7.9 | Other ** |
| *Archaeorhizomyces* sp. 5208_0 | A | MH248043 | 100 | 2.4 | 0.3 | 18.4 | 3.3 | 3.6 | Unknown |
| *Rhinocladiella* sp. 5208_3 | A | KM056296 | 98 | 1.7 | 5.4 | 0.01 | 0.5 | 2.0 | Saprotroph |
| *Arachnopeziza* sp. 5208_27 | A | MH558278 | 97 | 0.7 | 6.0 | - | 0.5 | 1.9 | Saprotroph |
| *Trichomerium* sp. 5208_23 | A | NR137946 | 97 | 2.7 | 3.9 | - | 0.4 | 1.9 | Endophyte |
| *Phacidium lacerum* | A | MN588163 | 100 | 0.6 | 0.3 | 0.2 | 3.7 | 1.7 | Pathogen |
| Unidentified sp. 5208_1 | A | MT595563 | 92 | 1.2 | 3.7 | 0.03 | 1.0 | 1.6 | Unknown |

**Table 3.** *Cont.*

| Fungal OTUs | Phylum * | Genbank Reference | Similarity, % | Needles, % | Shoots, % | Roots, % | Soil, % | All, % | Ecological Role |
|---|---|---|---|---|---|---|---|---|---|
| *Dothideomycetes* sp. 5208_5 | A | KX908472 | 99 | 0.6 | 0.5 | 0.1 | 3.3 | 1,5 | Unknown |
| *Cladosporium herbarum* | A | MT635288 | 100 | 2.1 | 0.9 | 0.3 | 1.7 | 1.5 | Saprotroph |
| *Cyphellophora sessilis* | A | KP400571 | 100 | 3.8 | 1.4 | 0.01 | 0.1 | 1.4 | Pathogen |
| *Rhizosphaera kalkhoffii* | A | KY003236 | 100 | 3.5 | 1.1 | 0.07 | 0.5 | 1.4 | Pathogen |
| *Lophium arboricola* | A | MK159395 | 100 | 0.2 | 4.6 | 0.01 | 0.3 | 1.4 | Unknown |
| *Chaetothyriales* sp. 5208_15 | A | KP400572 | 100 | 1.6 | 3.0 | 0.01 | 0.4 | 1.3 | Unknown |
| *Sporidesmium* sp. 5208_41 | A | MT596057 | 100 | 2.1 | 2.8 | 0.01 | 0.1 | 1.3 | Saprotroph |
| *Agaricomycetes* sp. 5208_12 | A | FJ553582 | 99 | 0.1 | 0.01 | 0.01 | 3.4 | 1.3 | Unknown |
| *Phialocephala fortinii* | A | MN947395 | 100 | 0.4 | 0.04 | 9.8 | 0.5 | 1.2 | Endophyte |
| *Microsphaeropsis olivacea* | A | MT561396 | 100 | 0.6 | 1.5 | 0.1 | 1.5 | 1.1 | Other ** |
| *Malassezia restricta* | B | LT854697 | 100 | 1.0 | 0.6 | 0.4 | 1.7 | 1.1 | Other ** |
| *Clavulina* sp. 5208_24 | B | OU498806 | 99 | - | - | - | 2.9 | 1.1 | Mycorrhizal |
| *Chaetothyriales* sp. 5208_2 | A | JQ342183 | 99 | 0.2 | 0.4 | 0.04 | 2.4 | 1.1 | Unkonwn |
| *Pezizomycotina* sp. 5208_65 | A | KP843512 | 96 | 0.4 | 3.1 | - | 0.3 | 1.0 | Unkonwn |
| *Mycena cinerella* | B | KT900146 | 100 | 0.1 | 0.01 | 9.5 | 0.4 | 0.9 | Saprotroph |
| *Blumeria graminis* f. sp. *tritici* | A | MT162615 | 100 | 0.01 | 0.002 | - | 2.4 | 0.9 | Pathogen |
| *Trechispora* sp. 5208_19 | B | JX392812 | 99 | 0.002 | - | 9.4 | 0.1 | 0.8 | Saprotroph |
| *Umbelopsis dimorpha* | Z | MT138616 | 100 | 0.2 | 0.01 | 0.01 | 2.1 | 0.8 | Endophyte |
| Total of 25 OTUs, % | | | | 54.9 | 39.4 | 48.4 | 33.3 | 42.1 | |

* A—Ascomycota, B—Basidiomycota, Z—Zygomycota. ** Unrelated to plants.

The relative abundance of most common plant pathogenic fungal OTUs is shown in Table 4. Plant pathogens were found to be most abundant in needles (9.7%) and soil (6.7%), when in shoots (4.6%), and these were least common in the roots (0.2%). In needles, the most common plant pathogens were *C. sessilis* (3.8%) and *R. kalkhoffii* (3.5%), in the soil—*P. lacerum* (3.8%) and *Microsphaeropsis olivacea* (1.5%), in shoots—*M. olivacea* (1.5%) and *C. sessilis* (1.4%), and in roots—*Tapesia lividofusca* (0.15%) and *Neonectria* sp. 5208_421 (0.15%) (Table 4).

**Table 4.** Occurrence and relative abundance of the 15 most common plant pathogenic, mycorrhizal and endophytic fungi (shown as a proportion of all high-quality fungal sequences) found in different substrates of *Picea abies*. The data from different sites are combined.

| Fungi | Phylum * | Genbank Reference | Similarity, % | Needles, % | Shoots, % | Roots, % | Soil, % | All, % |
|---|---|---|---|---|---|---|---|---|
| Plant pathogens | | | | | | | | |
| *Phacidium lacerum* | A | MN588163 | 100 | 0.6 | 0.3 | 0.2 | 3.7 | 1.7 |
| *Cyphellophora sessilis* | A | KP400571 | 100 | 3.8 | 1.4 | 0.01 | 0.1 | 1.4 |
| *Rhizosphaera kalkhoffii* | A | MN547387 | 100 | 3.5 | 1.1 | 0.07 | 0.5 | 1.4 |
| *Microsphaeropsis olivacea* | A | MT561396 | 100 | 0.6 | 1.5 | 0.1 | 1.5 | 1.2 |
| *Exobasidium arescens* | B | FJ896135 | 99 | 0.8 | 0.1 | - | 0.04 | 0.3 |
| *Coniochaeta hoffmannii* | A | MN341268 | 100 | 0.00 | 0.8 | - | 0.03 | 0.2 |
| *Phaeomoniella pinifoliorum* | A | MK762595 | 100 | - | - | - | 0.5 | 0.2 |
| *Alternaria infectoria* | A | MT635276 | 100 | 0.2 | 0.03 | - | 0.1 | 0.1 |
| *Hendersonia pinicola* | A | KT000192 | 100 | 0.05 | 0.03 | 0.01 | 0.2 | 0.1 |
| *Typhula* sp. 5208_156 | B | MN902561 | 95 | 0.07 | 0.2 | 0.1 | 0.03 | 0.07 |
| *Neonectria* sp. 5208_421 | A | LR603781 | 100 | - | - | 0.1 | 0.1 | 0.05 |
| *Alternaria alternata* | A | OL636518 | 100 | 0.1 | 0.03 | 0.1 | 0.04 | 0.05 |
| *Ganoderma applanatum* | B | MN906143 | 100 | 0.1 | 0.06 | 0.02 | 0.01 | 0.03 |
| *Fusarium* sp. 5208_517 | A | MT557415 | 99 | 0.01 | 0.01 | - | 0.06 | 0.03 |
| *Hymenoscyphus fraxineus* | A | MT155386 | 100 | 0.01 | 0.01 | 0.02 | 0.04 | 0.02 |

**Table 4.** *Cont.*

| Fungi | Phylum * | Genbank Reference | Similarity, % | Needles, % | Shoots, % | Roots, % | Soil, % | All, % |
|---|---|---|---|---|---|---|---|---|
| Total of 15 plant pathogens, % | | | | 9.8 | 5.6 | 0.7 | 6.9 | 6.9 |
| Mycorrhizal | | | | | | | | |
| *Clavulina* sp. 5208_24 | B | OU498806 | 99 | - | - | - | 2.9 | 1.1 |
| *Inocybe nitidiuscula* | B | AM882913 | 99 | - | - | - | 1.9 | 0.7 |
| *Cenococcum geophilum* | A | HM189724 | 100 | 0.02 ** | 0.003 ** | 2.5 | 0.8 | 0.5 |
| *Piloderma lanatum* | B | KP783452 | 100 | - | 0.002 ** | 0.005 | 1.2 | 0.5 |
| *Inocybe geophylla* | B | MK961172 | 99 | - | - | - | 1.2 | 0.4 |
| *Russula firmula* | B | DQ422017 | 100 | - | - | 0.03 | 1.2 | 0.4 |
| *Amphinema* sp. 5208_105 | B | KP125811 | 100 | - | - | - | 0.9 | 0.3 |
| *Lactarius rufus* | B | MK838331 | 100 | 0.03 ** | 0.002 ** | 2.9 | 0.06 | 0.3 |
| *Tricholoma* sp. 5208_110 | B | MK607553 | 100 | - | - | - | 0.5 | 0.2 |
| *Inocybe geophylla* | B | MT594793 | 100 | - | - | 0.005 | 0.5 | 0.2 |
| *Tylospora asterophora* | B | MG597438 | 100 | - | - | 0.04 | 0.4 | 0.2 |
| *Amphinema* sp. 5208_188 | B | MF352678 | 99 | - | - | - | 0.4 | 0.1 |
| *Cortinarius scotoides* | B | MW555551 | 99 | - | - | - | 0.3 | 0.1 |
| *Piloderma sphaerosporum* | B | MK131527 | 100 | 0.05 ** | - | 0.2 | 0.2 | 0.1 |
| *Inocybe pseudodestricta* | B | JF908157 | 100 | - | - | - | 0.3 | 0.1 |
| Total of 15 mycorrhizal OTUs, % | | | | 0.1 | 0.007 | 5.8 | 12.7 | 5.4 |
| Endophytes | | | | | | | | |
| *Trichomerium* sp. 5208_23 | A | NR_137946 | 97 | 2.7 | 3.9 | - | 0.4 | 1.9 |
| *Phialocephala fortinii* | A | MN947395 | 100 | 0.4 | 0.0 | 9.8 | 0.5 | 1.2 |
| *Umbelopsis dimorpha* | Z | MT138616 | 100 | 0.2 | 0.005 | 0.01 | 2.1 | 0.8 |
| *Mollisia scopiformis* | A | OM337553 | 100 | 0.003 | 0.6 | - | 0.05 | 0.2 |
| *Mollisia* sp. 5208_387 | A | OK430930 | 97 | 0.003 | - | 0.7 | 0.007 | 0.06 |
| *Mollisia* sp. 5208_1830 | A | MG195564 | 96 | 0.003 | 0.002 | 0.03 | 0.002 | 0.005 |
| *Mollisia novobrunsvicensis* | A | MT026439 | 100 | 0.002 | 0.007 | - | - | 0.002 |
| *Vestigium* sp. 5208_802 | A | NR_121556 | 94 | - | 0.003 | - | 0.003 | 0.002 |
| *Mollisia fusca* | A | LC425049 | 98 | 0.005 | 0.002 | - | - | 0.002 |
| *Phialocephala fusca* | A | KU668953 | 99 | - | - | - | 0.004 | 0.002 |
| *Phialocephala* sp. 5208_2177 | A | AB671500 | 98 | - | - | - | 0.004 | 0.002 |
| *Phialocephala bamuru* | A | MN006138 | 97 | - | - | 0.02 | - | 0.002 |
| *Mollisia* sp. 5208_3430 | A | MG195527 | 98 | - | - | 0.02 | - | 0.002 |
| *Mycroceros* sp. 5208_4863 | B | KT186373 | 96 | - | 0.007 | - | - | 0.002 |
| *Cadophora* sp. 5208_2138 | A | KY987540 | 97 | 0.002 | - | - | 0.002 | 0.001 |
| Total of 15 endophyte OTUs, % | | | | 3.4 | 4.5 | 10.5 | 3.0 | 4.2 |

* A—Ascomycota, B—Basidiomycota, Z—Zygomycota. ** Likely present as spores.

The relative abundance of fungal functional groups in different substrates and sites is shown in Figure 7. Among the identified functional groups, the most abundant were saprotrophs: in roots these composed 18.6%, in shoots—18.1%, in the soil—9.4%, and in needles—3.6% (Figure 7). The relative abundance of plant pathogenic fungi was 5.5% in needles, 5.3% in the soil, 5.3% in shoots, and 1.1% in roots. Among all samples and sites, 11.6% of fungal sequences were assigned to "others", which included fungi that are not associated with plants (mostly animal pathogens). In root and soil samples, the abundance of mycorrhizal fungi was 13.5% and 18.6%, respectively. The relative abundance of endophytes in roots was 10.5%, in the shoots—8.1%, in needles—4.3%, and soil—2.1% (Figure 7).

NMDS showed that fungal communities in different substrates of *P. abies* were largely different, and thus, substrate-specific ($p < 0.001$) (Figure 8). Analysis of the data with or without rare OTUs (<50 reads) did not have a larger effect on the output of NMDS as both of these were similar.

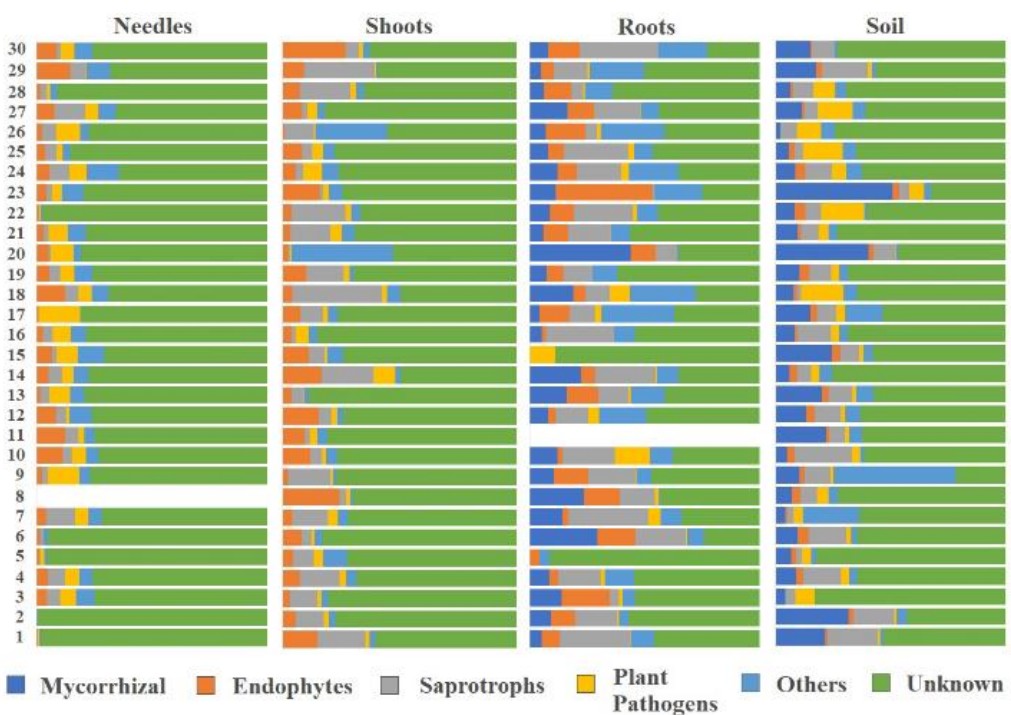

**Figure 7.** Relative abundance of fungal functional groups in different substrates (needles, shoots, roots, and the soil) of *Picea abies. Other* represent fungi (animal pathogens), which are not associated with plants. Site numbers to the left are as in Table 1 and Figure 1.

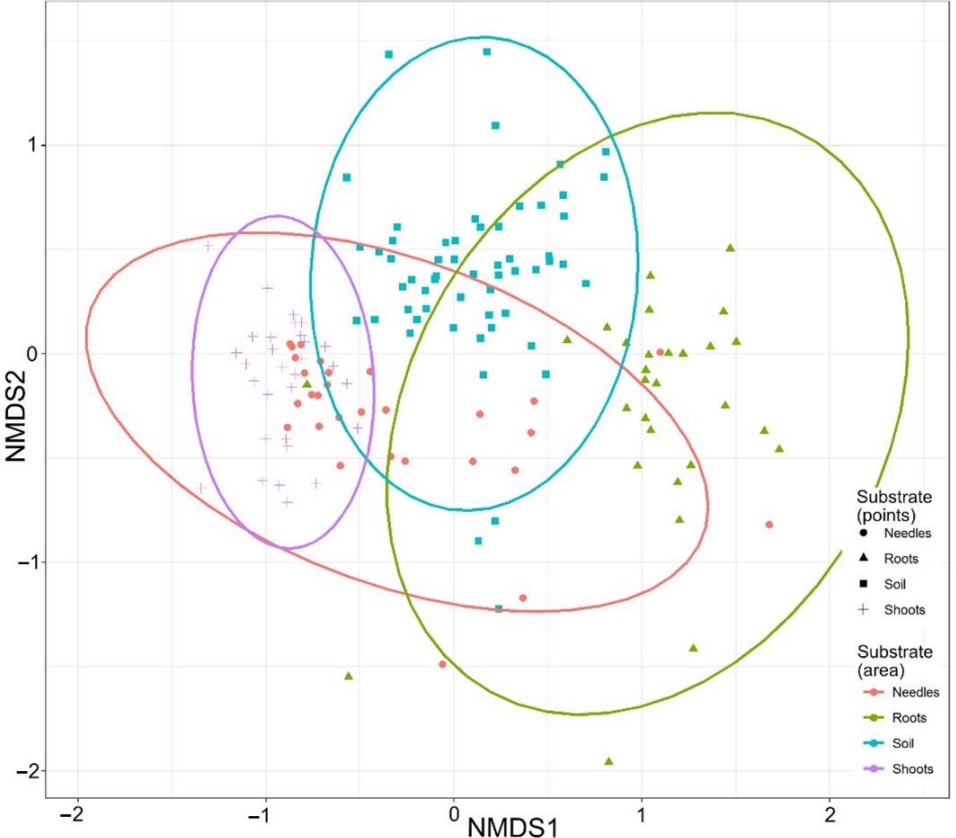

**Figure 8.** Nonmetric multidimensional scaling (NMDS) of fungal communities showing differences and similarities in needle, shoot, root, and the rhizosphere soil samples of *Picea abies*. For each substrate, each point in the diagram represents a single sampling site.

Furthermore, NMDS of fungal communities showed that soil physical parameters (moisture and fertility) did not have a significant effect on the composition of fungal communities in the soil ($p > 0.05$) (stress value: 0.17) (Figure 9A). However, soil moisture had a significant effect on the composition of fungal communities in needles, shoots, and roots of *P. abies* growing in wetlands (P) as compared to normal moisture (N) or temporary waterlogged soils (L) ($p < 0.05$) (Figure 9A). Moreover, soil fertility had a significant effect on the composition of fungal communities in needles and roots of *P. abies* growing in moderate fertility (c) soils as compared to high fertility (d) soils ($p < 0.05$) (Figure 9B). Stand age did not have a significant effect on the composition of fungal communities in needles, shoots, roots or the rhizosphere soil ($p < 0.05$) (Figure 9C). Although forest vegetation type did not have a significant effect on the composition of fungal communities in the soil ($p > 0.05$), some effects were observed for shoot and root samples. For example, in roots, significant differences in the composition of fungal communities were between the *Oxalidosa* (ox) vegetation type and *Hepatico-oxalidosa* (hox), *Filipendulo-mixtoherbosa* (fils), *Vacciniosa* (v), *Oxalido-nemorosa* (oxn) vegetation types ($p < 0.05$), and between *Myrtillo-oxalidosa* (mox) vegetation type and hox, fils, v and oxn vegetation types ($p < 0.05$) (Figure 9D). In shoots, significant differences in the composition of fungal communities were between *oxalidosa* (ox) vegetation type and hox, v, *Myrtillosa* (m) vegetation types ($p < 0.05$).

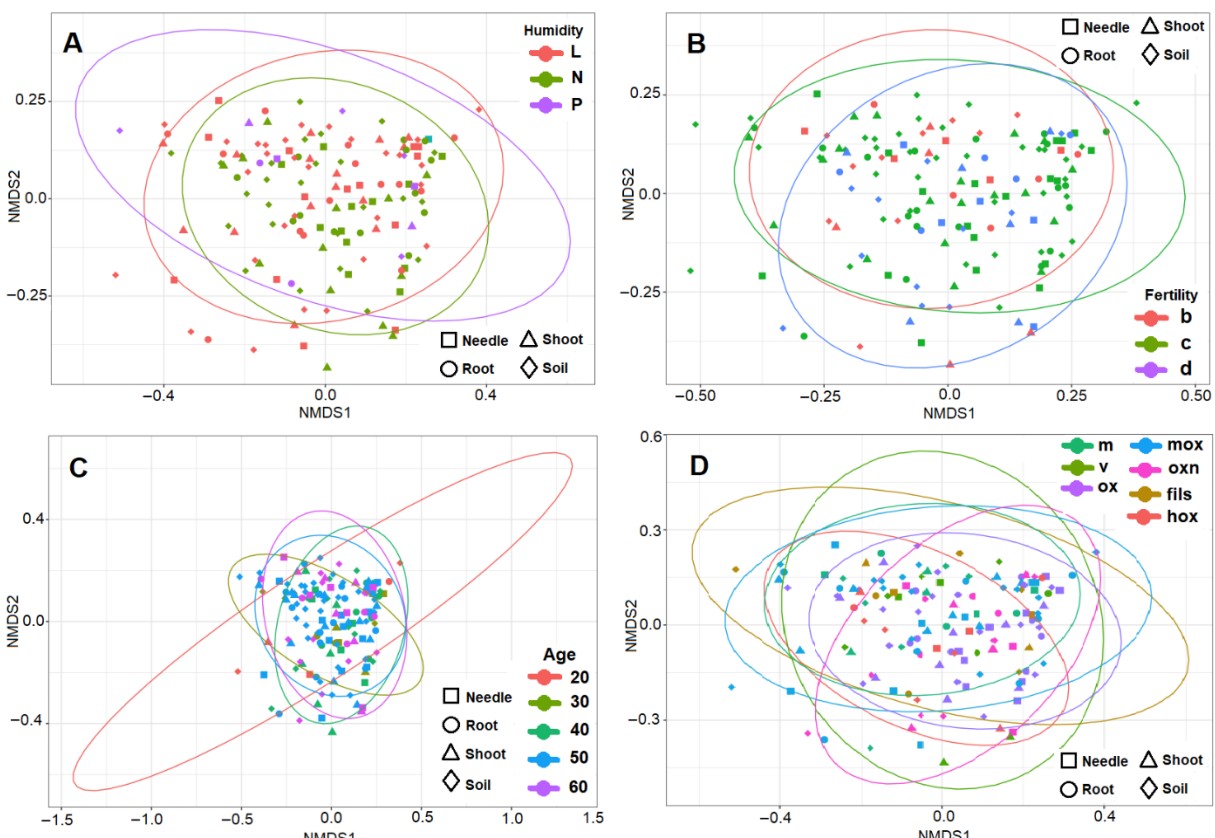

**Figure 9.** Nonmetric multidimensional scaling (NMDS) of fungal communities detected in needles, shoots, roots, and the rhizosphere soil of *Picea abies*. For each substrate, each point in the diagram represents a single sampling site. NMDS shows the impact of soil moisture (**A**), soil fertility (**B**), stand age (**C**), and vegetation type (**D**) on the composition of fungal communities.

The Sørensen similarity index of fungal communities ranged between 0.19 and 0.65 when the comparison was done among different substrates, i.e., shoots vs. needles—0.65, shoots vs. soil—0.39, shoots vs. roots—0.19, needles vs. soil—0.42, needles vs. roots—0.26 and soil vs. roots—0.34. In different substrates, the Shannon diversity index of fungal communities ranged between 1.68 and 3.29 in the roots, 0.14–4.36 in the needles, 2.53–4.34

in the shoots, and 2.92–5.01 in the soil (Table 2). The Mann–Whitney test showed that the Shannon diversity index of fungal communities differed significantly among different substrates ($p < 0.05$). An exception was the needle and shoot samples, which in this respect did not differ significantly from each other ($p > 0.05$).

## 4. Discussion

We studied fungal communities associated with the functional tissues (needles, shoots, and roots) and the rhizosphere soil of *P. abies*, which is one of most economically and ecologically important tree species in Central and Northern Europe as it occupies a large geographical area and grows under different environmental conditions [3]. As the present study encompassed several environmental conditions and habitats of *P. abies* (Table 1), it provided new and in-depth information on the diversity and composition of associated fungal communities in the region. It also revealed the potential effects of different factors on richness and composition of these fungal communities (Figures 4 and 9). The fungal species richness associated with *P. abies* was found to be high even though some site- and sample-specific variations were observed (Table 2). Interestingly, it was highest in the soil, then in the needles and shoots, and lowest in the roots (Table 2, Figure 2), showing the capacity of each habitat to support fungal diversity. This is not surprising as the fungal species richness in the soil was shown to be particularly high [57,58]. In general, soil fungi can occupy different ecological niches depending on available resources [59]. Organic and mineral nutrients present in the soil create favourable conditions for fungal activities such as decomposition and nutrient assimilation [60,61]. With a high diversity and complexity of fungal communities in the soil, the rate of decomposition and the release of nutrients increases [62], which also stimulates the uptake of nutrients by plants [63,64]. These factors promote tree viability and growth at the same time making them more tolerant to pathogens.

Although the majority of fungal OTUs were detected, even higher OUT richness could be revealed with deeper sequencing (Figure 2). Furthermore, in different sites and substrates, the detected diversity of fungal OTUs varied substantially (Figures 3 and 6), thereby highlighting the complexity of fungal colonisation patterns as well as interactions among the host trees, fungi, and local environmental conditions. Although the study included a number of different environmental conditions and habitats (Table 1), the comparison among different sites showed that within each substrate (needles, shoots, roots, and the soil), the richness of fungal OTUs was statistically similar (Figures 5 and 8, Table 2). The richness and composition of fungal communities are known to be affected by a variety of biotic and abiotic factors [65,66]. For example, the soil pH is one of the most important determinants of microbial communities in the soil as their richness and composition varies depending on the pH gradient [67,68]. This is in agreement with results of the present study as the richness of fungal species in the soil increased with the increase of soil pH (Figure 4A). The soil pH was also found to be among principal factors explaining ECM fungal diversity.

In many cases, $P_2O_5$, $K_2O$ and salts were commonly studied as these soil parameters are known to affect plant growth, but information on how these parameters affect the fungal species richness in forest soils is scarce. Therefore, the results of the present study provided new valuable information, namely how these parameters affect the fungal species richness associated with *P. abies*. Interestingly, the increasing $P_2O_5$ increased the fungal species richness in roots (Figure 4B). This may be due to the fact that phosphorus is essential for plant nutrition and growth, which may increase the allocation of carbohydrates belowground, thereby benefiting root-associated microorganisms [69]. The increasing amounts of $K_2O$ had the most pronounced effect on the richness of fungal species in the needles (Figure 4C) as $K_2O$ is essential for photosynthesis, including such functions as reduced respiration and energy losses, and enhanced translocation of sugars and starch [70].

The tree age had a positive effect of the fungal species richness in roots but not in other substrates. Fungi associated with plant roots were shown to be dynamic throughout plant

life [71,72]. As the plant develops and matures, the morphology and development of roots change, creating new spaces for the emergence and distribution of mycobiota. Therefore, there should be more different niches suitable for the fungi to establish in the root system of the older plants than in the younger ones [73]. By contrast, the study showed that in needles, the fungal species richness is highest in young and most actively growing trees (Figure 4E). In contrast to our expectations, the tree age had little or no effect on the richness of fungal OTUs, including ECM species, in the soil.

The increase of tree defoliation promoted the fungal species richness in needles and soil, but adversely affected these in shoots and roots (Figure 4D). In agreement, it was shown that defoliated trees often have a lower diversity of ECM fungi in the roots and a higher diversity and relative abundance of saprotrophic and pathogenic fungi in the soil [74]. This may be due to the fact that many saprotrophic fungi may feed on dead mycorrhizal structures, but may also benefit from dead organic matter such as dying/dead roots [75]. Changes in soil fungal community in defoliated stands may be due to the fall of dead needles on the ground, what may constitute a new substrate for colonisation, thereby indirectly affect fungal communities [76,77]. The reason for the observed defoliation was not established, but can possibly be due to insect attacks, diseases or climatic factors [38,78]. It was shown that damages caused by insects in the phyllosphere can invoke drastic alteration in fungal communities associated with this habitat [38].

The average annual temperature and annual precipitation were shown to be useful indicators of plant and animal diversity [79]. In the present study, both of these environmental factors had a similar effect on the fungal species richness in roots and needles, but the effect of these factors was different in shoots and in the soil (Figure 4G,H). It was shown before that temperature and precipitation may have a different effect on fungal diversity in different parts of plants [80,81]. Together with the soil pH, the annual precipitation was found to be among major factors describing fungal diversity in the soil, at the same time, demonstrating that the lack of precipitation may result not only in the decline of host trees but also in the decline of fungal diversity.

In the present study, NMDS showed that fungal communities were best explained by the substrate (needles, shoots, roots, and the soil) (Figure 8). NMDS also showed that both the forest vegetation type and soil physical properties did not have a significant effect on the composition of soil fungal communities (Figure 9A) even though small differences in soil physical properties can be expected to impact soil fungal communities [82,83]. Moreover, it was shown that well-aerated soils have generally a higher diversity of microorganisms than waterlogged ones [84]. The possibility should not be excluded that *P. abies* as a dominant tree species, had a major (homogenising) impact on fungal communities in the soil, leading to more similar soil fungal communities at different study sites. By contrast, certain vegetation types and soil physical properties had a significant effect on fungal communities associated with functional tissues of *P. abies* (Figure 9B,D). These effects can probably be explained by differences in water and nutrient availability, and thus, differences in nutrition of *P. abies*, which may determine the abundance and composition of associated fungal communities in specific tissues. In roots, these effects were most pronounced for mycorrhizal fungi, while in the aboveground parts—for endophytic fungi (Figure 7). In agreement, it was shown that the abundance of mycorrhizal fungi is affected by different soil properties, the time of sampling and climatic conditions [85,86]. Interactions between trees and fungal endophytes and patterns of colonisation are still not well understood, but generally these fungi are ubiquitous [33]. Although endophytic fungi may colonise tissues without causing symptoms [39,87], these may include different functional groups of fungi such as symbionts, latent pathogens, or saprotrophs [39,88].

Several different fungal phyla were detected among which Ascomycota and Basidiomycota were most common. Ascomycota was found to be more common in the aboveground tissues (needles and shoots), while Basidiomycota—belowground (in roots and soil), even though the difference was not significant. The phylum Ascomycota is the largest in the fungal kingdom, its species has a broad distribution and adapted to a variety of

habitats [89,90]. A higher abundance of Basidiomycota belowground can be attributed to the presence of basidiomycetous mycorrhizal and saprotrophic fungi, which play key roles in nutrient recycling in forest ecosystems [91,92].

The study also revealed that different tissues and the rhizosphere soil of *P. abies* were inhabited by a number of plant pathogenic fungi, but their relative abundance was rather low (Figure 7, Table 4), indicating that at the time of sampling, they did not cause any significant damage to the trees. The dominant pathogenic fungi were often substrate-specific (Table 4). In needles, the most abundant pathogens were *C. sessilis*, *R. kalkhoffi* and *E. arescens*, which are widespread in Europe. *Cyphellophora sessilis* can cause characteristic symptoms known as black sooty mould disease, which can occur on needles and shoots of different tree species [38,93,94]. The fungus has a negative effect on the tree's respiration process and appears to benefit from other tree damages [38]. Interestingly, the occurrence of *C. sessilis* negatively correlated with the stand age. The abundance of *R. kalkhoffi* and *E. arescens* was found to be slightly higher than in other similar studies [38,95]. As these pathogens are often associated with older needles, the sampling strategy, i.e., the use of two-year-old needles, may have contributed to their higher abundances [95–97]. *Microsphaeropsis olivacea* (syn. *Coniothyrium olivaceum*) was the most common pathogen in shoots (Table 4). Although it can occur as an endophyte [98,99], it was also shown to cause brown spine rot in colonised tree tissues [100]. It has a broad host range and geographical distribution, suggesting that it can adapt to a variety of conditions. The predominant establishment in asymptomatic *P. abies* shoots may suggest that under appropriate conditions, e.g., when trees are stressed, it may become pathogenic, develop rapidly, and cause the disease. Plant pathogenic fungus *P. lacerum* was among the most common fungi in the soil (Tables 3 and 4). Although it was suggested to be a widespread endophyte [101], it was also shown to be a weak pathogen of *P. sylvestris* and *Juniperus* needles [102]. However, the negative effect of *P. lacerum* on *P. abies* was not shown before. In the present study, *P. lacerum* showed a strong positive correlation with $P_2O_2$, Ca, and Mg, and a strong negative correlation with an average annual temperature. Pathogenic fungi from the genera *Alternaria* and *Fusarium* were also among most dominant in the soil (Table 3). These are generally known as soil-borne pathogens, which are commonly found in soils of the temperate climate zone [103,104]. Although their relative abundance was relatively low, climate change and higher temperatures in the soil can be expected to favour the activity and spread of these fungi [105]. Pathogenic fungi were least common in the roots and included representatives from genera such as *Neonectria*, *Hymenoscyphus*, *Alternaria* or *Ganoderma* (Table 4). Although many of these are generalists and commonly found in tree roots [106], *H. fraxineus* is a pathogen of *Fraxinus* spp. in Europe and is not associated with *P. abies*. The detection of *H. fraxineus* in different samples of the study was probably due to the presence of its propagules on the surface of different tissues (needles, shoots, or roots) and in the soil as the surface of our samples was not sterilised, and the disease caused by this fungus is active in the area [107]. The soil and tree roots were commonly colonised by mycorrhizal fungi (Table 4, Figure 7), which may also have limited the occurrence and activity of pathogenic fungi [108]. The mechanism for this was suggested to be the secretion of antimicrobial compounds and/or the completion for the space and resources. Moreover, mycorrhizal colonisation may lead to improved tree health due to enhanced nutrition, resulting in higher overall disease tolerance [109,110].

## 5. Conclusions

The functional tissues and rhizosphere soil of *P. abies* were inhabited by a species-rich communities of fungi. Within each substrate, fungal communities appeared to be similar, but several environmental variables had a significant effect on their diversity and community composition. The latter may suggest that fungi in different functional tissues and the rhizosphere soil of *P. abies* can be affected by climate change to a different extent with consequences for forest health and sustainability. The continuous monitoring of

fungal diversity and community composition is needed to better understand the short- and long-term effects of climate change in forest ecosystems.

**Supplementary Materials:** The following are available online at https://www.mdpi.com/article/10.3390/f13071103/s1, Table S1: Relative abundance (%) of fungal OTUs associated with needles, shoots, roots, and the rhizosphere soil of *Picea abies* in Lithuania.

**Author Contributions:** Conceptualization, A.M. (Audrius Menkis); methodology, A.M. (Adas Marčiulynas), D.M., A.G., V.M. and A.M. (Audrius Menkis); software, V.M. and I.F.; validation, A.M. (Adas Marčiulynas), D.M. and A.M. (Audrius Menkis); formal analysis, A.M. (Adas Marčiulynas) and I.F; investigation, D.M., J.L., A.G. and V.M.; writing—original draft preparation, A.M. (Adas Marčiulynas); writing—review and editing, D.M., J.L. and A.M. (Audrius Menkis); visualization, D.M. and J.L. All authors have read and agreed to the published version of the manuscript.

**Funding:** This project has received funding from European Social Fund (project No. 09.3.3-LMT-K-712-01-0039) under grant agreement with the Research Council of Lithuania (LMTLT).

**Institutional Review Board Statement:** Not applicable.

**Informed Consent Statement:** Not applicable.

**Data Availability Statement:** Not applicable.

**Conflicts of Interest:** The authors declare no conflict of interest.

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
