# Peer review of "High Variability of Fungal Communities Associated with the Functional Tissues and Rhizosphere Soil of Picea abies in the Southern Baltics"

_forests, doi:10.3390/f13071103_

Round 1

Reviewer 1 Report

Review of Forests MS High stochasticity of fungal communities associated with the 2 functional tissues and rhizosphere soil of Picea abies in the 3 southern Baltics

This manuscript describes a study looking at the fungal communities (of all functional groups) associated with spruce (Picea abies) across numerous sites in Lithuania (LT). The methods and results of the study are technically strong but the manuscript and study objectives are lacking in focus that can help the reader see what is so interesting about this study.  Further, there are no clearly stated hypotheses that place the study in bigger conceptual questions.  As written, the manuscript is largely a verbatim report on the data recovered from the different tissues of a spruce – a catalog of Lithuanian fungi.  Although important as general knowledge, it doesn’t address (in current form) novel and compelling ideas or problems. I would encourage a reworking of the overall manuscript to build a more interesting presentation of perhaps more novel directions.  One direction might be to dig deeper into the idea of stochasticity and what that means and what specific hypotheses could be tested with these data regarding stochasticity? Alternatively, could these data examine questions about specific pathogens and symbionts? Could these data test if certain pathogens “avoid” certain symbionts? Another direction could be to split the paper into two separate papers (root/soil fungi for one; shoot/needle for another) and address more functionally relevant questions – e.g. the trend towards greater root fungi in older forests is interesting given what we know about early stage vs later stage ectomycorrhizal fungi.  In summary, the paper shows patterns that are pretty well known.  For instance, I’m not surprised that there are no mycorrhizal fungi in the shoots and needles. It should be refocused to be more compelling.

Here are some line by line comments to help the rewrite:

Line 26: Should be “the most common fungi based on sequence read abundance”.  The relationship between sequence reads and actual biological abundance/biomass is not direct.  You can rely on it for presence-absence, but not abundance.

Line 35: Overall about the Introduction: Could the Intro be shortened?  Seems to ramble, perhaps each paragraph could be a third shorter and more concise.  Guide the intro towards tangible, testable hypotheses. 

Lines 103-105: This is an ok objective, but I wonder how novel or interesting it is. Could the ms be focused on specific hypotheses?  Perhaps state based on principles that soil will have greater diversity than other parts because soil accumulates most of the propagules and is a longer lived reservoir of DNA? Related to this, what drove the selection of sites and could that be used as a motivation to guide thinking about specific hypotheses? 

Line 114: The sampling of LT is great! But why this set of sites specifically instead of others?  Is this related to the Level I grid?  Explain what this is.  Does this include similar stand ages, size (ha), stem densities, dbh, management practices? All of these could be relevant factors to fungal diversity beyond the types of tissues from which the study sampled.

Line 145: In Table 1, the number of sample columns (the final set of columns) are all exactly the same, no need to include beyond a brief statement in the methods where this table is described. Further, somewhere the forest vegetation typology/classification system used here should be explained in more detail.  Probably good to do this in the Methods. 

Table 1 caption:  “N” = I think should probably be “moisture” or “mesic” rather than “humidity”.  Elsewhere in the ms, “humidity” should be replace by “moisture”. 

Line 186:  Where was the PacBio sequencing performed?

Table 2: Should it be “No. of sequences/fungal OTU” instead?  Using “taxa” can be confusing as that could lump into a broader taxonomic categories.  But, OTU is a specific to a single entity as defined in the methods. 

Figure 3: This is an interesting figure but I’m still not sure if does the best job of showing the results intended.  Perhaps a more concise version with one map of LT but with bar graphs at each point for the relative fungi would be better?  Not sure.  

Author Response

This manuscript describes a study looking at the fungal communities (of all functional groups) associated with spruce (Picea abies) across numerous sites in Lithuania (LT). The methods and results of the study are technically strong but the manuscript and study objectives are lacking in focus that can help the reader see what is so interesting about this study.  Further, there are no clearly stated hypotheses that place the study in bigger conceptual questions.  As written, the manuscript is largely a verbatim report on the data recovered from the different tissues of a spruce – a catalog of Lithuanian fungi.  Although important as general knowledge, it doesn’t address (in current form) novel and compelling ideas or problems. I would encourage a reworking of the overall manuscript to build a more interesting presentation of perhaps more novel directions.  One direction might be to dig deeper into the idea of stochasticity and what that means and what specific hypotheses could be tested with these data regarding stochasticity? Alternatively, could these data examine questions about specific pathogens and symbionts? Could these data test if certain pathogens “avoid” certain symbionts? Another direction could be to split the paper into two separate papers (root/soil fungi for one; shoot/needle for another) and address more functionally relevant questions – e.g. the trend towards greater root fungi in older forests is interesting given what we know about early stage vs later stage ectomycorrhizal fungi.  In summary, the paper shows patterns that are pretty well known.  For instance, I’m not surprised that there are no mycorrhizal fungi in the shoots and needles. It should be refocused to be more compelling.

Response: the aim of the study was to provide a comprehensive overview on fungal communities associated with Picea abies. To our best knowledge, there are no similar studies in the geographical area and/or on P. abies. However, we agree that the hypothesis was missing, but it is included now.

We appreciate specific suggestions on how the manuscript could be reworked. However, we consider that we do provide examples and discuss the stochasticity. Please also observe that we do examine common pathogens, mycorrhizal fungi and endophytes (Table 4), and discuss their significance to growth and health of forest trees.

We would like to avoid splitting the paper into the two as suggested. We consider (as also indicated in the Introduction) that the strength of the manuscript is that different samples (needles, shoots, roots, and the soil) were analysed together, providing a more complete picture on fungal biodiversity associated with P. abies.

In fact, we do detect in low quantities mycorrhizal fungi on shoots and needles, but interpret this as the presence of spores.

Here are some line by line comments to help the rewrite:

Line 26: Should be “the most common fungi based on sequence read abundance”.  The relationship between sequence reads and actual biological abundance/biomass is not direct.  You can rely on it for presence-absence, but not abundance.

Response: changed as suggested

Line 35: Overall about the Introduction: Could the Intro be shortened?  Seems to ramble, perhaps each paragraph could be a third shorter and more concise.  Guide the intro towards tangible, testable hypotheses.

Response: we consider that the Introduction is already concise i.e., only 1.5 of the page. Shortening would cut out some important background information. However, we removed some text and added a hypothesis.

Lines 103-105: This is an ok objective, but I wonder how novel or interesting it is. Could the ms be focused on specific hypotheses?  Perhaps state based on principles that soil will have greater diversity than other parts because soil accumulates most of the propagules and is a longer lived reservoir of DNA? Related to this, what drove the selection of sites and could that be used as a motivation to guide thinking about specific hypotheses?

Response: the aim is also explained by the two following sentences, thereby putting the study in a broader context of climate change, tree health and effects of fungal communities. The hypothesis was also added. The sites were selected to represent different P. abies stands and habitats across the country.

Line 114: The sampling of LT is great! But why this set of sites specifically instead of others?  Is this related to the Level I grid?  Explain what this is.  Does this include similar stand ages, size (ha), stem densities, dbh, management practices? All of these could be relevant factors to fungal diversity beyond the types of tissues from which the study sampled.

Response: thank you for this observation. Additional information was added to Materials and Methods.

Line 145: In Table 1, the number of sample columns (the final set of columns) are all exactly the same, no need to include beyond a brief statement in the methods where this table is described. Further, somewhere the forest vegetation typology/classification system used here should be explained in more detail.  Probably good to do this in the Methods.

Response: as suggested, columns with samples were removed. Additional information on classification was added to Materials and Methods.

Table 1 caption:  “N” = I think should probably be “moisture” or “mesic” rather than “humidity”.  Elsewhere in the ms, “humidity” should be replace by “moisture”.

Response: replaced as suggested

Line 186:  Where was the PacBio sequencing performed?

Response: It was clarified that the PacBio sequencing was carried out at the SciLifeLab in Uppsala, Sweden.

Table 2: Should it be “No. of sequences/fungal OTU” instead?  Using “taxa” can be confusing as that could lump into a broader taxonomic categories.  But, OTU is a specific to a single entity as defined in the methods.

Response: as suggested taxa were replaced by OUTs.

Figure 3: This is an interesting figure but I’m still not sure if does the best job of showing the results intended.  Perhaps a more concise version with one map of LT but with bar graphs at each point for the relative fungi would be better?  Not sure. 

Response: we consider that the figure provides a good overview of fungal richness in different samples. Combining all data into one figure would make it overcrowded and hardly readable. Therefore, we would like to keep this figure intact.

Reviewer 2 Report

This manuscript focused on the fungal community variety of different functional tissues and rhizosphere soil of Picea abies in the southern Baltics. Its a valuable study. The writing style and English language is fine. As a whole, the aim, method and results of this experiment was illustrated clearly. However, there are some minor comments.

Detailed comments:

In general, please write the genus name and species name of plant and fungi in italic type of whole manuscript. ( marked with yellow).

The data analysis werent deep and need to be dug deeper.

-line 3: theres no author belongs to the third department among the authors listed. 

-line 19: It looks the word size P. abies is bigger than others?

-line 67: It's well known about the fungi promote the nutrient and carbon cycles. But water is seldom reported. However, the water cycle was mainly caused by plant. Please check the references carefully.

-line 108-110: What's the reason about the territory of Lithuania to be outside the range of P. abies? Please give a brief explanation and cite a reference.

-line 123-126: The method of collecting rhizosphere soil isnt reasonable. Rhizosphere soil, should be next to the root of plant. They should be sampled with a brush gently from the root. The description of sampling soil in this manuscript is the method of the whole forest soil collecting.  

-line 263-265: Fig 4A showed the changing trend of species richness with pH. There are some question. Is the pH value of different functional tissues and soil can reach to 2? Or the abscissa wasnt the pH value? However, its possible that the pH value equal to 2 about the needle and shoot, but not the soil, especially the soil samples is the forest basic soil not real rhizosphere soil depending the sampling method described in 2.1. It will be very hard for plant to grow. Theres a suggestion to showed the physicochemical property in text with a table.

-line 319: 25 most common fungi were listed in table 3, including their fungal sequences and proportion. However, these dominant species were the major components and have important function in the fungal community. Their ecological function should be showed here. The similarity of Cladophialophora sp. 5208_1 with MT595563 was only 92% and couldnt be considered as same genus. This question also be found in table 4, Typhula sp. 5208_156.

-line 343: From Fig 7, we found the important different function group in different tissues. But only dominant plant pathogenic fungi were listed in table 4. However, mycorrhizal fungi and endophytes showed their inhabiting preference and the dominant species of these functional group need be listed.

-line 366: 4 results of NMDS didnt show the difference of fungal community with different environmental factors. However, these figures didnt showed the discrepancy of different tissues. Maybe theres some problems with data processing method. Some OTUs with very low abundance (<50) disturb the results. Can try to remove them and redo. In addition, its not enough judge the environmental factors effecting the fungal community or not, CCA need to analysis. Please add this.

Author Response

This manuscript focused on the fungal community variety of different functional tissues and rhizosphere soil of Picea abies in the southern Baltics. It’s a valuable study. The writing style and English language is fine. As a whole, the aim, method and results of this experiment was illustrated clearly. However, there are some minor comments.

Response: we acknowledge these observations and are grateful for valuable comments and suggestions.

Detailed comments:

In general, please write the genus name and species name of plant and fungi in italic type of whole manuscript. ( marked with yellow).

Response: changed to italic.

The data analysis weren’t deep and need to be dug deeper.

Response: as suggested we did additional analyses, i.e. added information on ecological roles to Table 3, expanded Table 4 by adding most common mycorrhizal and endophytic fungi, and included Figure 8.   

-line 3: there’s no author belongs to the third department among the authors listed.

Response: corrected

-line 19: It looks the word size “P. abies” is bigger than others?

Response: corrected

-line 67: It's well known about the fungi promote the nutrient and carbon cycles. But water is seldom reported. However, the water cycle was mainly caused by plant. Please check the references carefully.

Response: “water” was excluded

-line 108-110: What's the reason about the territory of Lithuania to be outside the range of P. abies? Please give a brief explanation and cite a reference.

Response: it was specified that it is due to the climate change. References were added.

-line 123-126: The method of collecting rhizosphere soil isn’t reasonable. Rhizosphere soil, should be next to the root of plant. They should be sampled with a brush gently from the root. The description of sampling soil in this manuscript is the method of the whole forest soil collecting. 

Response: it is now specified that soil samples were taken in the vicinity of P. abies trees.

-line 263-265: Fig 4A showed the changing trend of species richness with pH. There are some question. Is the pH value of different functional tissues and soil can reach to 2? Or the abscissa wasn’t the pH value? However, it’s possible that the pH value equal to 2 about the needle and shoot, but not the soil, especially the soil samples is the forest basic soil not real rhizosphere soil depending the sampling method described in 2.1. It will be very hard for plant to grow. There’s a suggestion to showed the physicochemical property in text with a table.

Response: the lowest pH value (2.4) was detected in the peatland, site no. 3. Peatland soils are generally known to have a low pH value. As suggested, the Table 1 was expanded adding additional stand and soil chemical data.

-line 319: 25 most common fungi were listed in table 3, including their fungal sequences and proportion. However, these dominant species were the major components and have important function in the fungal community. Their ecological function should be showed here. The similarity of Cladophialophora sp. 5208_1 with MT595563 was only 92% and couldn’t be considered as same genus. This question also be found in table 4, Typhula sp. 5208_156.

Response: Information on functional groups was added to the Table 3. The sequence similarity was checked, and the species name was corrected.

-line 343: From Fig 7, we found the important different function group in different tissues. But only dominant plant pathogenic fungi were listed in table 4. However, mycorrhizal fungi and endophytes showed their inhabiting preference and the dominant species of these functional group need be listed.

Response: Table 4 was expanded adding mycorrhizal fungi and endophytes.

-line 366: 4 results of NMDS didn’t show the difference of fungal community with different environmental factors. However, these figures didn’t showed the discrepancy of different tissues. Maybe there’s some problems with data processing method. Some OTUs with very low abundance (<50) disturb the results. Can try to remove them and redo. In addition, it’s not enough judge the environmental factors effecting the fungal community or not, CCA need to analysis. Please add this.

Response: We have redone NMDS analyses with excluded OUTs that have very low abundance data (<50), but we got very similar result. However, the analysis of different substrates (needles, shoots, roots, and the soil) showed that fungal communities in these differ significantly. This NMDS plot is now included in the manuscript as Fig. 8. 

Round 2

Reviewer 1 Report

Review of second draft:

Some minimal edits have improved this manuscript a bit, but the deeper structural issues remain. I strongly encourage the authors revisit the motivation and identify their hypotheses in more detail and depth. This will help put their abundant (and important) results into a stronger, more adequate context.

Stochasticity has a fairly precise definition: patterns best described by random chance.  In a study that aims to test whether a pattern is stochastic (see line 125), the measured pattern should be compared to what would be expected in a null model that is randomly generated.  So, in order to test the hypothesis that diversity across the sites is stochastic, the observed diversity should be compared to a null model (see for example Ning et al 2019 https://www.pnas.org/doi/10.1073/pnas.1904623116. ) So, if a primary objective of the authors is to test this hypothesis, the measured patterns need to be compared against a random null model and a test of this comparison made. As is, I don’t see this in the study. Further, I wonder how to do this given that the samples here are taken from across the different tissues of spruce trees sampled across the entire country of LT? If this hypothesis remains an objective, one suggestion is to generate a random model for each tissue type, ie, needles, shoots, roots, soils and then compare the diversity of each tissue type against the random expectation.  

Alternatively, change the title and refocus the aim of the paper away from “stochasticity” per se. Describing these patterns as highly “variable” (thus the title could be “High variability…”) might be better as that doesn’t imply an underlying random or deterministic process. 

I still have concerns about the rationale of the study.  Is the main goal to examine diversity across sites? Further, what is the underlying reason why it is thought that diversity should be variable?  For example, I could see arguing that sites that were older and had more tree density (and thus root density) would likely have more ectomycorrhizal fungi.  Moreover, why wouldn't the composition also follow suit?   What about the diversity within a site across the different tissues?  Is this not of interest?  I think the hypotheses can be expanded and more clearly described for each tissue type across the sites and how the explanatory factors included could explain these.  For example, an argument could be made for EcM fungi that more fertile sites may have fewer fungi since EcM are considered to help in nutrient acquisition (the logic follows then that in more fertile soils the spruce would not need EcM fungi).  Similarly, some EcM fungi prefer nutrient rich soils (e.g. nitrophilic Russula spp), so one could propose that composition should shift accordingly.  What are the hypotheses and predictions for the other tissue types across the variables measured? As is, I see only that the objective was to simply measure all the fungi of the different tissue types across the spruce of LT; and without any in depth explanation as what they might expect, predict, hypothesize and why they would think so.

For answers to my questions I looked at the statistical methods the authors used.  I see that in addition to general descriptive statistics of the diversity patterns, on lines 271-273 they used glm to examine how “substrate and environmental variables” impact OTU richness.  Further, in the results, I see that the authors state (line 330) “species richness was significantly affected by variables reflecting the soil chemistry.” And that “stand age” as well as many other variables were significant predictors of fungal communities in different “substrates”.  But, then I see in the Discussion, lines 502-503 that “the diversity was largely stochastic”.  This contradicts the results as, again, if it were stochastic, it would be random and unrelated to variables like soil chemistry and stand age.  Across sites the diversity is variable, true, but the glm shows that there are factors that can explain this diversity pattern, at least in part. 

Author Response

Some minimal edits have improved this manuscript a bit, but the deeper structural issues remain. I strongly encourage the authors revisit the motivation and identify their hypotheses in more detail and depth. This will help put their abundant (and important) results into a stronger, more adequate context.

Response: we are grateful for valuable suggestions and guidance, which helped to improve the manuscript.

Stochasticity has a fairly precise definition: patterns best described by random chance.  In a study that aims to test whether a pattern is stochastic (see line 125), the measured pattern should be compared to what would be expected in a null model that is randomly generated.  So, in order to test the hypothesis that diversity across the sites is stochastic, the observed diversity should be compared to a null model (see for example Ning et al 2019 https://www.pnas.org/doi/10.1073/pnas.1904623116. ) So, if a primary objective of the authors is to test this hypothesis, the measured patterns need to be compared against a random null model and a test of this comparison made. As is, I don’t see this in the study. Further, I wonder how to do this given that the samples here are taken from across the different tissues of spruce trees sampled across the entire country of LT? If this hypothesis remains an objective, one suggestion is to generate a random model for each tissue type, ie, needles, shoots, roots, soils and then compare the diversity of each tissue type against the random expectation.  

Response: the hypothesis was refocused.

Alternatively, change the title and refocus the aim of the paper away from “stochasticity” per se. Describing these patterns as highly “variable” (thus the title could be “High variability…”) might be better as that doesn’t imply an underlying random or deterministic process. 

Response: the suggestion was adopted in the manuscript.

I still have concerns about the rationale of the study.  Is the main goal to examine diversity across sites? Further, what is the underlying reason why it is thought that diversity should be variable?  For example, I could see arguing that sites that were older and had more tree density (and thus root density) would likely have more ectomycorrhizal fungi.  Moreover, why wouldn't the composition also follow suit?   What about the diversity within a site across the different tissues?  Is this not of interest?  I think the hypotheses can be expanded and more clearly described for each tissue type across the sites and how the explanatory factors included could explain these.  For example, an argument could be made for EcM fungi that more fertile sites may have fewer fungi since EcM are considered to help in nutrient acquisition (the logic follows then that in more fertile soils the spruce would not need EcM fungi).  Similarly, some EcM fungi prefer nutrient rich soils (e.g. nitrophilic Russula spp), so one could propose that composition should shift accordingly.  What are the hypotheses and predictions for the other tissue types across the variables measured? As is, I see only that the objective was to simply measure all the fungi of the different tissue types across the spruce of LT; and without any in depth explanation as what they might expect, predict, hypothesize and why they would think so.

Response: the rationale was to examine whether fungal diversity differ across the sites and how it is affected by different variables to better understand if climate change can affect fungal communities at different sites in a similar or different way.

In the Introduction l. 112-120, we have now provided the reasoning why diversity should be variable across the sites and in different tissues. We do agree that older sites have the potential to accumulate a higher diversity, including ECM fungi.  

For answers to my questions I looked at the statistical methods the authors used.  I see that in addition to general descriptive statistics of the diversity patterns, on lines 271-273 they used glm to examine how “substrate and environmental variables” impact OTU richness.  Further, in the results, I see that the authors state (line 330) “species richness was significantly affected by variables reflecting the soil chemistry.” And that “stand age” as well as many other variables were significant predictors of fungal communities in different “substrates”.  But, then I see in the Discussion, lines 502-503 that “the diversity was largely stochastic”.  This contradicts the results as, again, if it were stochastic, it would be random and unrelated to variables like soil chemistry and stand age.  Across sites the diversity is variable, true, but the glm shows that there are factors that can explain this diversity pattern, at least in part. 

Response: Thank you for this observation. We agree that there was a contradiction, which is now resolved.
